

# Symbolic model checking quantum circuits in Maude

Canh Minh Do and Kazuhiro Ogata

School of Information Science, Japan Advanced Institute of Science and Technology, Asahidai, Nomi, Ishikawa, Japan

## ABSTRACT

This article presents a symbolic approach to model checking quantum circuits using a set of laws from quantum mechanics and basic matrix operations with Dirac notation. We use Maude, a high-level specification/programming language based on rewriting logic, to implement our symbolic approach. As case studies, we use the approach to formally specify several quantum communication protocols in the early work of quantum communication and formally verify their correctness: Superdense Coding, Quantum Teleportation, Quantum Secret Sharing, Entanglement Swapping, Quantum Gate Teleportation, Two Mirror-image Teleportation, and Quantum Network Coding. We demonstrate that our approach/implementation can be a first step toward a general framework to formally specify and verify quantum circuits in Maude. The proposed way to formally specify a quantum circuit makes it possible to describe the quantum circuit in Maude such that the formal specification can be regarded as a series of quantum gate/measurement applications. Once a quantum circuit has been formally specified in the proposed way together with an initial state and a desired property expressed in linear temporal logic (LTL), the proposed model checking technique utilizes a built-in Maude LTL model checker to automatically conduct formal verification that the quantum circuit enjoys the property starting from the initial state.

## INTRODUCTION

Quantum computing is a rapidly emerging technology that uses the laws of quantum mechanics to solve complex problems that are very hard for classical computers, such as discrete logarithms and factoring. Several quantum algorithms have been proposed showing a significant improvement over classical algorithms, such as the fast algorithms for discrete logarithms and factoring proposed by *Shor (1994)*. It is well known that cryptosystems relying on the hardness of discrete logarithms and factoring will be broken by large-scale quantum computers running Shor's fast algorithm in the future. Then, quantum communication involving quantum cryptography has attracted much attention from both industry and academia because it provides an efficient and highly secure communication channel relying on quantum mechanics phenomena, such as superposition, entanglement, and probabilistic measurement.

Corresponding author
Canh Minh Do, canhdo@jaist.ac.jp

Quantum circuits are a model of quantum computation, comprising a sequence of quantum gates, measurements, initializations of qubits, and possibly other actions. Quantum gates operate on quantum bits (qubits), the quantum counterpart of classical bits, and manipulate the state of a quantum system to perform quantum computations. The outputs of quantum circuits are quantum states, which can be measured to obtain classical outcomes with probabilities from which other actions can take place. Quantum circuits play a crucial role in the development of quantum algorithms because they are used to design and implement quantum algorithms before actually running on quantum computers. Because quantum computing is counter-intuitive and radically different from classical computing, the likelihood of errors in quantum algorithms and circuits is much higher than in classical algorithms. Therefore, it is critical to verify that quantum circuits (or algorithms) enjoy desired properties.

Model checking is a formal verification technique widely used in both academia and industry to systematically verify that systems satisfy desired properties. Quantum programs and quantum circuits are related concepts, but they differ in their level of abstraction and the way they represent quantum computations. Quantum circuits are low-level representations of quantum computation that can be used to implement quantum programs, while quantum programs are higher-level representations of quantum computations that can be expressed in a quantum programming language consisting of a series of instructions, especially the loop instruction. Although there are some model checkers dedicated to quantum programs, such as *Gay, Nagarajan & Papanikolaou (2008)*, *Feng, Yu & Ying (2013)* and *Feng et al. (2015)* (see *Ying & Feng (2018)*, *Ying & Feng (2021)*, *Turrini (2022)* for more details), there is still a gap between model checking quantum programs and quantum circuits due to different representations and no iteration in quantum circuits, which should be filled in. Moreover, because the verification of classical circuits using model checking has been proven to be a tremendously successful technique, model checking that quantum circuits satisfy desired properties would be a promising approach. There is a symbolic approach proposed by *Shi et al. (2021)* to (semi-)automatically reasoning about quantum circuits in Coq (https://coq.inria.fr/), an interactive theorem prover, but it often requires human users to provide necessary lemmas to complete its proofs.

This article presents a symbolic approach to model checking quantum circuits using a set of laws from quantum mechanics and basic matrix operations with Dirac notation proposed by *Dirac (1939)*. Concretely, quantum states, quantum gates, and measurements are described in Dirac notation instead of using explicitly complex vectors and matrices as proposed by *Paykin, Rand & Zdancewic (2017)*, making our representations more compact. Using the set of laws, we can systematically reason about the evolution of quantum states. We use Maude introduced by *Clavel et al. (2007)*, a high-level specification/programming language based on rewriting logic presented by *Meseguer (2012)*, to specify quantum states, some basic quantum gates (*e.g.*, Hadamard gate, controlled-NOT gate, and Pauli gates), and measurements on a standard basis with Dirac notation. Maude is equipped with a linear temporal logic (LTL) model checker and its reflective programming (or meta-programming) facilities have been used to develop several software tools, such as Maude-NPA introduced by *Escobar, Meadows & Meseguer (2007)*, its parallel version

developed by *Do et al. (2022)*, and a toolset of some parallel versions of the LTL model checker presented by *Do et al. (2021)*, *Do, Phyo & Ogata (2022)*, *Do et al. (2023)* and *Phyo et al. (2023)*. Therefore, Maude makes it possible/convenient to implement our idea and carry out case studies. This is why we adopt Maude for the research described in the article.

As case studies, we focus on using our approach to formally specify several quantum communication protocols in the early work of quantum communication and formally verify their correctness: Superdense Coding introduced by *Bennett & Wiesner (1992)*, Quantum Teleportation presented by *Bennett et al. (1993)*, Quantum Secret Sharing developed by *Hillery, Bužek & Berthiaume (1999)*, Entanglement Swapping proposed by *Zukowski et al. (1993)*, Quantum Gate Teleportation suggested by *Gottesman & Chuang (1999)*, Two Mirror-image Teleportation devised by *Williams (2008)*, and Quantum Network Coding originated by *Satoh, Gall & Imai (2012)*. In this article, we use eventual properties, a class of liveness properties, to express the desired properties for these quantum communication protocols. In addition to the desired properties, any properties that can be expressed in the scope of LTL can essentially be verified using the Maude LTL model checker with our approach. In this article, we do not directly tackle quantum circuits for complicated quantum algorithms, such as *Shor (1994)* and *Grover (1996)* because necessary quantum gates have not been developed yet and our symbolic reasoning for complex numbers is not sufficient to describe and reason about their behaviors adequately. Therefore, extending our approach to handle these algorithms would require further research, which would be one piece of our future work. Our specification is specifically tailored to quantum circuits, abstracting away from the details of concurrency and communication. To handle quantum cryptography, such as BB84 introduced by *Bennett & Brassard (2014)* and B91 introduced by *Ekert (1991)*, we need to be able to express concurrency and communication among participants in quantum protocols in our specification. Therefore, extending our approach to handle such quantum protocols would require further research, which would be one piece of our future work.

We demonstrate that our approach/implementation can be a first step toward a general framework to formally specify and verify quantum circuits. The proposed way to formally specify a quantum circuit makes it possible to describe the quantum circuit in Maude such that the formal specification can be regarded as a series of quantum gate/measurement applications. Once a quantum circuit has been formally specified in the proposed way together with an initial state and a desired property expressed in LTL, the proposed model checking technique utilizes a built-in Maude LTL model checker to automatically conduct formal verification that the quantum circuit enjoys the property starting from the initial state. Moreover, our specification considers the probabilities from measurements in quantum computation based on which the probability of a computation occurring is accumulated across states and so we are able to analyze both the quantitative and qualitative properties[1] of several quantum communication protocols with the built-in LTL model checker in Maude. Our implementation is publicly available at https://doi.org/10.5281/zenodo.10783951.

The present article is an extended and improved version of our conference article presented by *Do & Ogata (2023)* with some improvements as follows:

[1]Quantitative properties involve numerical aspects, such as probabilities or quantitative measures of system behavior, while qualitative properties focus on the presence or absence of certain behaviors. For example, the probability of each computation of a model is only zero or one that can be considered as qualitative properties, while the probability of each computation of a model is greater or less than a certain number other than zero or one that can be considered as quantitative properties.

- We fully support Pauli gates in our specification and some additional gates, including *S*, *T*, *CY*, *CZ*, *SWAP*, *CCY*, *CCZ*, and *CSWAP* gates. Moreover, the symbolic reasoning is refined and improved in order to conduct more case studies.
- We verify some more quantum communication protocols: Superdense Coding, Quantum Secret Sharing, Entanglement Swapping, Quantum Gate Teleportation, Two Mirror-image Teleportation, and Quantum Network Coding in order to demonstrate the usefulness of our approach for formally specifying and verifying quantum circuits in Maude.
- We identify that the original version of Quantum Gate Teleportation does not satisfy its desired property using our approach and support tool. We then propose a revised version of the protocol and verify that the revised one satisfies its desired property using our approach and support tool.
- Lastly, we describe how we specify complex numbers in Maude to symbolically reason on complex numbers with rational numbers for our case studies.

The rest of the article is organized as follows: 'Preliminaries' explains basic quantum mechanics and Kripke structures; 'Rewriting Logic and Quantum Circuits' explains how we can associate a rewrite theory with a quantum circuit *via* a Kripke structure; 'Symbolic Reasoning' describes how to construct terms and use a set of laws from quantum mechanics and matrix operations for symbolic reasoning using our approach; 'Formal Specification' details how to specify qubits, gates, measurements, and then quantum circuits in order to symbolically model check quantum circuits in a generic way; 'Symbolic Model Checking' demonstrates how to use our symbolic approach to model checking several quantum communication protocols in depth; 'Remark on Quantum Gate Teleportation' provides a remark on Quantum Gate Teleportation; 'Experimental Results' presents our experimental results; 'Discussion' discusses our limitations, some challenges in using the Maude LTL model checker, a classical model checker, to verify quantum circuits, and how we address them in this article; 'Related Work' reviews some existing work; and 'Conclusion' concludes the article with some pieces of future work.

# PRELIMINARIES

This section briefly describes some basic notations from quantum mechanics based on linear algebra (refer to *Nielsen & Chuang (2010)* for more details) and Kripke structures.

## Basic quantum mechanics

This section describes basic quantum mechanics based on the linear algebra approach. In classical computing, the fundamental unit of information is a bit whose value is either 0 or 1. In quantum computing, the counterpart is a *quantum bit* or *qubit*, which has two basis states, conventionally written in Dirac notation proposed by *Dirac (1939)* as $|\mathbf{0}\rangle$ and $|\mathbf{1}\rangle$, which denote two column vectors $\begin{pmatrix} 1 \\ 0 \end{pmatrix}$ and $\begin{pmatrix} 0 \\ 1 \end{pmatrix}$, respectively. In quantum theory, a general state of a quantum system is a superposition or linear combination of basis states. A single qubit has state $|\psi\rangle = \alpha|\mathbf{0}\rangle + \beta|\mathbf{1}\rangle$, where $\alpha$ and $\beta$ are complex numbers such that $|\alpha|^2 + |\beta|^2 = 1$. States can be represented by column complex vectors as follows:

**Table 1  Quantum gates by names, circuit forms, and the corresponding unitary matrices.**

| Operator | Gate | Matrix |
|---|---|---|
| Identity ($I_2$) | $-\boxed{I}-$ | $\begin{pmatrix} 1 & 0 \\ 0 & 1 \end{pmatrix}$ |
| Pauli-X ($X$) | $-\boxed{X}-$ | $\begin{pmatrix} 0 & 1 \\ 1 & 0 \end{pmatrix}$ |
| Pauli-Y ($Y$) | $-\boxed{Y}-$ | $\begin{pmatrix} 0 & -i \\ i & 0 \end{pmatrix}$ |
| Pauli-Z ($Z$) | $-\boxed{Z}-$ | $\begin{pmatrix} 1 & 0 \\ 0 & -1 \end{pmatrix}$ |
| Hadamard ($H$) | $-\boxed{H}-$ | $\frac{1}{\sqrt{2}}\begin{pmatrix} 1 & 1 \\ 1 & -1 \end{pmatrix}$ |
| Controlled-NOT ($CX$) (the first and second wires denote the control and target qubits, respectively) |  | $\begin{pmatrix} 1 & 0 & 0 & 0 \\ 0 & 1 & 0 & 0 \\ 0 & 0 & 0 & 1 \\ 0 & 0 & 1 & 0 \end{pmatrix}$ |

$$|\psi\rangle = \begin{pmatrix} \alpha \\ \beta \end{pmatrix} = \alpha|0\rangle + \beta|1\rangle,$$

where $\{|0\rangle, |1\rangle\}$ forms an orthonormal basis of the two-dimensional complex vector space. Formally, a quantum state is a unit vector in a Hilbert space $\mathcal{H}$, which is equipped with an inner product satisfying some axioms.

The basis $\{|0\rangle, |1\rangle\}$ is called the *standard* basis. Besides, we have some other bases of interest, such as the *diagonal* (or *dual*, or *Hadamard*) basis consisting of the following vectors:

$$|+\rangle = \frac{1}{\sqrt{2}}(|0\rangle + |1\rangle) \text{ and } |-\rangle = \frac{1}{\sqrt{2}}(|0\rangle - |1\rangle).$$

The evolution of a closed quantum system can be performed by a unitary transformation. If the state of a qubit is represented by a column vector, then a unitary transformation can be represented by a complex-value matrix $U$ such that $UU^\dagger = U^\dagger U = I$ or $U^\dagger = U^{-1}$, where $U^\dagger$ is the conjugate transpose of $U$. $U$ acts on the Hilbert space $\mathcal{H}$ transforming a state $|\psi\rangle$ to a state $|\psi'\rangle$ by a matrix multiplication such that $|\psi'\rangle = U|\psi\rangle$. There are some common quantum gates: the identity gate $I$, the Pauli gates $X$, $Y$, and $Z$, the Hadamard gate $H$, and the controlled-NOT gate $CX$. Note that the $CX$ gate performs on two qubits, while the remaining gates perform on a single qubit.

For example, the Hadamard gate on a single qubit performs the mapping $|0\rangle \mapsto \frac{1}{\sqrt{2}}(|0\rangle + |1\rangle)$ and $|1\rangle \mapsto \frac{1}{\sqrt{2}}(|0\rangle - |1\rangle)$. The controlled-NOT gate on pairs of qubits performs the mapping $|00\rangle \mapsto |00\rangle, |01\rangle \mapsto |01\rangle, |10\rangle \mapsto |11\rangle, |11\rangle \mapsto |10\rangle$, which can be understood as inverting the second qubit (referred to as the *target*) if and only if the first qubit (referred to as the *control*) is $1$. The common quantum gates are shown in Table 1 by names, circuit forms, and matrix representations, where $i$ is the imaginary unit.

A quantum measurement is described as a collection $\{M_m\}$ of measurement operators, where the indices $m$ refer to the measurement outcomes. It is required that the measurement

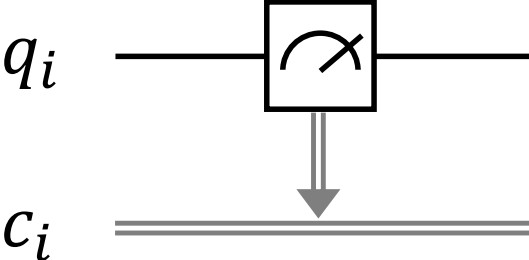

**Figure 1** The circuit form of the binary projective measurement, where the measurement outcome of the qubit $q_i$ is stored in the classical bit $c_i$.

operators satisfy $\sum_m M_m^\dagger M_m = I_{\mathcal{H}}$. If the state of a quantum system is $|\psi\rangle$ before the measurement, then the probability for the result $m$ is as follows:

$p(m) = \langle\psi|M_m^\dagger M_m|\psi\rangle$,

where $\langle\psi|$ is the dual of $|\psi\rangle$ such that $\langle\psi|^\dagger = |\psi\rangle$ and $|\psi\rangle^\dagger = \langle\psi|$. The state of the quantum system after the measurement is $\frac{M_m|\psi\rangle}{\sqrt{p(m)}}$ provided that $p(m) > 0$. For example, if a qubit is in state $\alpha|0\rangle + \beta|1\rangle$ and measuring with $\{M_0, M_1\}$ operators, we have the result 0 with probability $|\alpha|^2$ at the post-measurement state $|0\rangle$ and the result 1 with probability $|\beta|^2$ at the post-measurement state $|1\rangle$, where $M_0 = |0\rangle \times \langle 0|$ and $M_1 = |1\rangle \times \langle 1|$. The quantum measurement with $\{M_0, M_1\}$ operators is called the binary projective measurement. In this study, we only use the binary projective measurement and its circuit form is depicted in Fig. 1 as follows:

For multiple qubits, we use the tensor product of Hilbert spaces. Let $\mathcal{H}_1$ and $\mathcal{H}_2$ be two Hilbert spaces. Their tensor product $\mathcal{H}_1 \otimes \mathcal{H}_2$ is defined as a vector space consisting of linear combinations of the vectors $|\psi_1\psi_2\rangle = |\psi_1\rangle|\psi_2\rangle = |\psi_1\rangle \otimes |\psi_2\rangle$, where $|\psi_1\rangle \in \mathcal{H}_1$ and $|\psi_2\rangle \in \mathcal{H}_2$. Systems of two or more qubits may be in *entangled* states, meaning that states of qubits are correlated and inseparable. For example, we consider a measurement of the first qubit of the entangled state $\frac{1}{\sqrt{2}}(|00\rangle + |11\rangle)$. The result is either 0 with probability $\frac{1}{2}$ leaving its state $|00\rangle$ or 1 with probability $\frac{1}{2}$ leaving its state $|11\rangle$. In either case, a subsequent measurement of the second qubit gives a non-probabilistic result, which is immediate to the result of the first measurement before. Entanglement shows that an entangled state of two qubits cannot be expressed as a tensor product of single-qubit states. We can use $H$ and $CX$ gates to create entangled states as follows: $CX((H \otimes I)|00\rangle) = \frac{1}{\sqrt{2}}(|00\rangle + |11\rangle)$.

## Kripke structures

A Kripke structure $K$ is a tuple $\langle S, I, T, A, L\rangle$ as represented by *Clarke et al. (2018)*, where $S$ is a set of states, $I \subseteq S$ is the set of initial states, $T \subseteq S \times S$ is a left-total binary relation over $S$, $A$ is a set of atomic propositions, and $L$ is a labeling function whose type is $S \to 2^A$. Each element $(s, s') \in T$ is called a state transition from $s$ to $s'$ and $T$ may be called the state transitions (with respect to $K$). For a state $s \in S$, $L(s)$ is the set of atomic propositions that hold in $s$. A path $\pi$ is an infinite sequence $s_0, \ldots, s_i, s_{i+1}, \ldots$ such that $s_i \in S$ and $(s_i, s_{i+1}) \in T$

for each $i$. We use the following notations for paths: $\pi^i \triangleq s_i, s_{i+1}, \ldots, \pi_i \triangleq s_0, \ldots, s_i, s_i, s_i, \ldots,$ $\pi(i) \triangleq s_i$, where $\triangleq$ is used as "be defined as." $\pi^i$ is obtained by deleting the first $i$ states $s_0, s_1, \ldots, s_{i-1}$ from $\pi$. $\pi_i$ is obtained by taking the first $i+1$ states $s_0, s_1, \ldots, s_{i-1}, s_i$ and adding $s_i$ unboundedly many times at the end. $\pi(i)$ is the $i$th state $s_i$. Let $\mathcal{P}$ be the set of all paths. $\pi$ is called a computation if $\pi(0) \in I$. Let $\mathcal{C}$ be the set of all computations.

The syntax of a formula $\varphi$ in LTL for $K$ is as follows:

$\varphi := \top \mid p \mid \neg\varphi \mid \varphi \wedge \varphi \mid \bigcirc\varphi \mid \varphi\,\mathcal{U}\,\varphi$

where $p \in A$, and $\bigcirc$ and $\mathcal{U}$ are called the next temporal connective and the until temporal connective, respectively. We introduce the eventual temporal abbreviation $\diamond$ which is defined as follows:

- $\diamond\varphi \triangleq \top\,\mathcal{U}\,\varphi$

This eventual temporal abbreviation is also used in *Clarke et al. (2018)*.

Let $\mathcal{F}$ be the set of all formulas in LTL for $K$. Given an arbitrary path $\pi \in \mathcal{P}$ of $K$ and an arbitrary LTL formula $\varphi \in \mathcal{F}$ of $K$, $K, \pi \models \varphi$ is inductively defined as follows:

- $K, \pi \models \top$
- $K, \pi \models p$ iff $p \in L(\pi(0))$
- $K, \pi \models \neg\varphi_1$ iff $K, \pi \not\models \varphi_1$
- $K, \pi \models \varphi_1 \wedge \varphi_2$ iff $K, \pi \models \varphi_1$ and $K, \pi \models \varphi_2$
- $K, \pi \models \bigcirc\varphi_1$ iff $K, \pi^1 \models \varphi_1$
- $K, \pi \models \varphi_1\,\mathcal{U}\,\varphi_2$ iff there exists a natural number $i$ such that $K, \pi^i \models \varphi_2$ and for all natural numbers $j < i$, $K, \pi^j \models \varphi_1$

where $\varphi_1$ and $\varphi_2$ are LTL formulas. Then, $K \models \varphi$ iff $K, \pi \models \varphi$ for each computation $\pi \in \mathcal{C}$ of $K$.

In this article, we refer to $\diamond\varphi$ as eventual properties, which informally state that something will eventually happen. Termination or halting is one important system requirement that many systems should satisfy and can be expressed in LTL as an eventual property. Moreover, we aim to verify whether quantum circuits satisfy certain desired properties where something good eventually happens. For example, a qubit at the final state for each possible execution path is the same as another qubit at the initial state with a non-zero probability. Therefore, it is worthwhile to use eventual properties to express desired properties for our case studies under verification. For more details, the reader is referred to 'Symbolic Model Checking' to see how we express the desired properties for our case studies as eventual properties.

## REWRITING LOGIC AND QUANTUM CIRCUITS

This section describes how we can associate a rewrite theory with a quantum circuit *via* a Kripke structure at a conceptual level, enabling the use of LTL model checking to verify that the quantum circuit enjoys a desired property.

A rewrite theory $\mathcal{R}$ is a triple $(\Sigma, E, R)$, where

- $\Sigma$ is an order-sorted signature consisting of a set of *sorts*, *subsorts*, and *function* symbols,

- $(\Sigma, E)$ forms an order-sorted equational theory with $E$ being a collection of (possibly conditional) *equations* $t = t'$,
- $R$ is a collection of (possibly conditional) *rewrite rules* $l \to r$.

Terms are built from variables, constants, and function symbols from $\Sigma$, and each term has a sort. The equations in $E$ are used to reduce a term into a normal form, while the rewrite rules in $R$ modulo $E$ are used to make local transitions in systems, making it possible to rewrite one term to another term. We can associate a Kripke structure $K = \langle S, I, T, A, L \rangle$ to a rewrite theory $\mathcal{R} = (\Sigma, E, R)$ as presented in *Clavel et al. (2007*, Chapter 13). In short, each term $t$ in $\mathcal{R}$ can be regarded as a state $s \in S$ in $K$; and each rewriting step from $t$ to $t'$ can be regarded as a state transition $(s, s') \in T$ in $K$, where $t$ and $t'$ are terms of the same sort with their corresponding states $s, s' \in S$. $A$ and $L$ are not necessary parts of $\mathcal{R}$ and can be specified later in terms of constants and equations, respectively, to determine whether atomic propositions are true at a given state.

A quantum circuit can be described as a series of applications of quantum gates, measurements, and conditional gates, which are applied based on the outcomes of measurements. The input of a quantum circuit is a quantum state, and so is the output. The input and the output of a quantum circuit can be regarded as the initial state and the final state belonging to $S$ in $K$, where the initial state also belongs to $I$ in $K$. Therefore, a quantum state can be specified as a term in the rewrite theory $\mathcal{R}$. The application of quantum gates manipulates a quantum state to perform quantum computation, which is specified in terms of equations in $\mathcal{R}$ so that we can reason about quantum computation. The application of a quantum gate can be regarded as a deterministic state transition in $K$ since it transforms a quantum state into another quantum state. As a result, the application of a quantum gate can be specified by a rewrite rule in $\mathcal{R}$. Besides quantum gates, we can conduct a measurement on a quantum state to obtain classical outcomes based on which other quantum actions (*e.g.*, quantum gates) can take place. As mentioned before, we are only interested in the binary projective measurement in this article. Therefore, the application of a measurement can be regarded as a non-deterministic state transition in $K$ since the measurement may make a quantum state collapse into one of two different possibilities of quantum states with probabilities. As a result, the application of a measurement can be specified by two rewrite rules in $\mathcal{R}$. Note that there are only two rewrite rules for quantum measurements, while there are as many rewrite rules as the number of quantum gates supported by the rewrite theory $\mathcal{R}$. If the equations in $\mathcal{R}$ are sufficient to reason about any quantum computation and the rewrite rules in $\mathcal{R}$ support sufficient quantum gates, the rewrite theory $\mathcal{R}$ can simulate the behavior of any quantum circuit, making it applicable in a generic sense.

Given a concrete quantum circuit described as a series of quantum gates, quantum measurement, and conditional gates, along with an initial quantum state, the rewrite theory $\mathcal{R}$ can simulate the behavior of the quantum circuit by addressing all possible execution paths starting from the initial quantum state. We can associate $K$ with $\mathcal{R}$ so as to conduct LTL model checking and verify that the quantum circuit satisfies a desired property. The desired property of the quantum circuit can be constructed based on the

atomic propositions (regarded as state predicates) within the scope of LTL language. For each possible execution path, we can examine each quantum state and check which atomic propositions hold at the state. Thus, LTL model checking can verify whether the quantum circuit satisfies the desired property.

# SYMBOLIC REASONING

This section introduces some terms used in our symbolic reasoning and a set of laws used to reduce terms. The "symbolic" word means that we use bra-ket notation, which means $\langle \psi |$ and $| \psi \rangle$, instead of explicitly complex vectors and matrices as proposed by *Paykin, Rand & Zdancewic (2017)*, which makes our representations more compact. Moreover, we can deal with not only concrete values but also symbolic values (representing arbitrary values) for complex numbers reasoning.

## Terms

Terms are built from scalars and basic vectors with some operations.

- Scalars are complex numbers. We extend rational numbers supported in Maude to deal with complex numbers. Some operations for scalars, such as multiplication, division, addition, conjugation, absolute, power, and square roots are specified. The reader who is interested in how to specify complex numbers in Maude is referred to Appendix A.
- Basic vectors are the ones of the standard basis written in Dirac notation as $|\mathbf{0}\rangle$ and $|\mathbf{1}\rangle$.
- Operations for matrices consist of scalar multiplication $\cdot$, matrix product $\times$, matrix addition $+$, tensor product $\otimes$, and the conjugate transpose $A^\dagger$ of a matrix $A$.

In Dirac notation, $\langle \mathbf{0} |$ is the dual of $|\mathbf{0}\rangle$ such that $\langle \mathbf{0} |^\dagger = |\mathbf{0}\rangle$ and $|\mathbf{0}\rangle^\dagger = \langle \mathbf{0} |$; similarly for $\langle \mathbf{1} |$. The terms $|\mathbf{j}\rangle \times \langle \mathbf{k} |$ and the inner product of ket vectors $|\mathbf{j}\rangle$ and $|\mathbf{k}\rangle$ may be written shortly as $|\mathbf{j}\rangle\langle \mathbf{k} |$ and $\langle \mathbf{j} | \mathbf{k}\rangle$ for any $\mathbf{j}, \mathbf{k} \in \{\mathbf{0}, \mathbf{1}\}$. By using these notations, we can intuitively explain how quantum operations work. For example, the $X$ gate performs mapping $|\mathbf{0}\rangle \mapsto |\mathbf{1}\rangle$ and $|\mathbf{1}\rangle \mapsto |\mathbf{0}\rangle$. Therefore, we specify the $X$ gate as $|\mathbf{0}\rangle\langle \mathbf{1} | + |\mathbf{1}\rangle\langle \mathbf{0} |$ in Maude instead of using explicitly the matrix representation $\begin{pmatrix} 0 & 1 \\ 1 & 0 \end{pmatrix}$. We have $X |\mathbf{0}\rangle = |\mathbf{1}\rangle\langle \mathbf{0} || \mathbf{0}\rangle + |\mathbf{0}\rangle\langle \mathbf{1} || \mathbf{0}\rangle = |\mathbf{1}\rangle$ because of laws L1 and L3 in Table 2 and similarly for $X |\mathbf{1}\rangle = |\mathbf{0}\rangle$.

We conventionally specify some basic matrices $B_i$ for $i \in [0..3]$ as follows:

$$B_0 = |\mathbf{0}\rangle \times \langle \mathbf{0} |, \qquad B_1 = |\mathbf{0}\rangle \times \langle \mathbf{1} |, \qquad B_2 = |\mathbf{1}\rangle \times \langle \mathbf{0} |, \qquad B_3 = |\mathbf{1}\rangle \times \langle \mathbf{1} |.$$

The $X$, $Y$, $Z$, $CX$, and $H$ gates are then a linear combination of the matrices $B_i$ as follows:

$$X = B_1 + B_2, \qquad Y = (-i) \cdot B_1 + i \cdot B_2, \qquad Z = B_1 + (-1) \cdot B_3,$$
$$CX = B_0 \otimes I_2 + B_3 \otimes X, \qquad H = \frac{1}{\sqrt{2}} \cdot B_0 + \frac{1}{\sqrt{2}} \cdot B_1 + \frac{1}{\sqrt{2}} \cdot B_2 + (-\frac{1}{\sqrt{2}}) \cdot B_3.$$

## Laws

We use a set of laws in Table 2 derived from the properties of quantum mechanics and basic matrix operations, and thus, they are immediately sound. The reader who is interested in their proofs in Coq is referred to *Shi et al. (2021)*. Because $|\mathbf{0}\rangle$ and $|\mathbf{1}\rangle$ can be viewed as $2 \times 1$ matrices, then the laws actually describe matrix calculations with Dirac notation, zero and identity matrices, and scalars. These laws are described by equations in Maude and

**Table 2   A set of laws used for symbolic reasoning.**

| No. | Law |
| --- | --- |
| L1 | $\langle \mathbf{0}\,|\,\mathbf{0}\,\rangle = \langle \mathbf{1}\,|\,\mathbf{1}\,\rangle = 1, \langle \mathbf{1}\,|\,\mathbf{0}\,\rangle = \langle \mathbf{0}\,|\,\mathbf{1}\,\rangle = 0$ |
| L2 | Associativity of $\times, +, \otimes$ and Commutativity of $+$ |
| L3 | $0 \cdot \boldsymbol{A}_{m \times n} = \mathbf{O}_{m \times n},\ c \cdot \mathbf{O} = \mathbf{O},\ 1 \cdot \boldsymbol{A} = \boldsymbol{A}$ |
| L4 | $c \cdot (\boldsymbol{A} + \boldsymbol{B}) = c \cdot \boldsymbol{A} + c \cdot \boldsymbol{B}$ |
| L5 | $c_1 \cdot \boldsymbol{A} + c_2 \cdot \boldsymbol{A} = (c_1 + c_2) \cdot \boldsymbol{A}$ |
| L6 | $c_1 \cdot (c_2 \cdot \boldsymbol{A}) = (c_1 \cdot c_2) \cdot \boldsymbol{A}$ |
| L7 | $(c_1 \cdot \boldsymbol{A}) \times (c_2 \cdot \boldsymbol{B}) = (c_1 \cdot c_2) \cdot (\boldsymbol{A} \times \boldsymbol{B})$ |
| L8 | $\boldsymbol{A} \times (c \cdot \boldsymbol{B}) = (c \cdot \boldsymbol{A}) \times \boldsymbol{B} = c \cdot (\boldsymbol{A} \times \boldsymbol{B})$ |
| L9 | $\boldsymbol{A} \otimes (c \cdot \boldsymbol{B}) = (c \cdot \boldsymbol{A}) \otimes \boldsymbol{B} = c \cdot (\boldsymbol{A} \otimes \boldsymbol{B})$ |
| L10 | $\mathbf{O}_{m \times n} \times \boldsymbol{A}_{n \times p} = \boldsymbol{A}_{m \times n} \times \mathbf{O}_{n \times p} = \mathbf{O}_{m \times p}$ |
| L11 | $\boldsymbol{I}_m \times \boldsymbol{A}_{m \times n} = \boldsymbol{A}_{m \times n} \times \boldsymbol{I}_n = \boldsymbol{A}_{m \times n}$ |
| L12 | $\boldsymbol{A} + \mathbf{O} = \mathbf{O} + \boldsymbol{A} = \mathbf{O}$ |
| L13 | $\mathbf{O}_{m \times n} \otimes \boldsymbol{A}_{p \times q} = \boldsymbol{A}_{p \times q} \otimes \mathbf{O}_{m \times n} = \mathbf{O}_{mp \times nq}$ |
| L14 | $\boldsymbol{A} \times (\boldsymbol{B} + \boldsymbol{C}) = \boldsymbol{A} \times \boldsymbol{B} + \boldsymbol{A} \times \boldsymbol{C}$ |
| L15 | $(\boldsymbol{A} + \boldsymbol{B}) \times \boldsymbol{C} = \boldsymbol{A} \times \boldsymbol{C} + \boldsymbol{B} \times \boldsymbol{C}$ |
| L16 | $(\boldsymbol{A} \otimes \boldsymbol{B}) \times (\boldsymbol{C} \otimes \boldsymbol{D}) = (\boldsymbol{A} \times \boldsymbol{C}) \otimes (\boldsymbol{B} \times \boldsymbol{D})$ |
| L17 | $\boldsymbol{A} \otimes (\boldsymbol{B} + \boldsymbol{C}) = \boldsymbol{A} \otimes \boldsymbol{B} + \boldsymbol{A} \otimes \boldsymbol{C}$ |
| L18 | $(\boldsymbol{A} + \boldsymbol{B}) \otimes \boldsymbol{C} = \boldsymbol{A} \otimes \boldsymbol{C} + \boldsymbol{B} \otimes \boldsymbol{C}$ |
| L19 | $(c \cdot \boldsymbol{A})^{\dagger} = c^* \cdot \boldsymbol{A}^{\dagger},\ (\boldsymbol{A} \times \boldsymbol{B})^{\dagger} = \boldsymbol{B}^{\dagger} \times \boldsymbol{A}^{\dagger}$ |
| L20 | $(\boldsymbol{A} + \boldsymbol{B})^{\dagger} = \boldsymbol{A}^{\dagger} + \boldsymbol{B}^{\dagger},\ (\boldsymbol{A} \otimes \boldsymbol{B})^{\dagger} = \boldsymbol{A}^{\dagger} \otimes \boldsymbol{B}^{\dagger}$ |
| L21 | $\boldsymbol{I}_m^{\dagger} = \boldsymbol{I}_m,\ \mathbf{O}_{m \times n}^{\dagger} = \mathbf{O}_{n \times m},\ (\boldsymbol{A}^{\dagger})^{\dagger} = \boldsymbol{A}$ |
| L22 | $|\mathbf{0}\rangle^{\dagger} = \langle \mathbf{0}|,\ \langle \mathbf{0}|^{\dagger} = |\mathbf{0}\rangle,\ |\mathbf{1}\rangle^{\dagger} = \langle \mathbf{1}|,\ \langle \mathbf{1}|^{\dagger} = |\mathbf{1}\rangle$ |

are used to automatically reduce terms until no more matrix operation is applicable. Some laws dedicated to simplifying the expressions about complex numbers are also specified in Maude by means of equations, but we do not mention them here for brevity.

For example, we would like to reduce the term $\boldsymbol{CX} \times ((\boldsymbol{H} \otimes \boldsymbol{I}) \times |\mathbf{0}\rangle \otimes |\mathbf{0}\rangle)$ to check whether its result is $\frac{1}{\sqrt{2}} \cdot |\mathbf{0}\rangle \otimes |\mathbf{0}\rangle + \frac{1}{\sqrt{2}} \cdot |\mathbf{1}\rangle \otimes |\mathbf{1}\rangle$. The term says that the $\boldsymbol{H}$ gate acts on the first qubit followed by the $\boldsymbol{CX}$ gate where the control and target bits are the first and second qubits, respectively. The simplification of the term goes as follows:

$\boldsymbol{H} \times |\mathbf{0}\rangle$

$$= (\frac{1}{\sqrt{2}} \cdot \boldsymbol{B}_0 + \frac{1}{\sqrt{2}} \cdot \boldsymbol{B}_1 + \frac{1}{\sqrt{2}} \cdot \boldsymbol{B}_2 + (-\frac{1}{\sqrt{2}}) \cdot \boldsymbol{B}_3) \times |\mathbf{0}\rangle \quad \text{(by replacement of } \boldsymbol{H})$$

$$= \frac{1}{\sqrt{2}} \cdot \boldsymbol{B}_0 \times |\mathbf{0}\rangle + \frac{1}{\sqrt{2}} \cdot \boldsymbol{B}_1 \times |\mathbf{0}\rangle + \frac{1}{\sqrt{2}} \cdot \boldsymbol{B}_2 \times |\mathbf{0}\rangle + (-\frac{1}{\sqrt{2}}) \cdot \boldsymbol{B}_3 \times |\mathbf{0}\rangle \quad \text{(by law L15)}$$

$$= \frac{1}{\sqrt{2}} \cdot |\mathbf{0}\rangle \times \langle \mathbf{0}| \times |\mathbf{0}\rangle + \frac{1}{\sqrt{2}} \cdot |\mathbf{0}\rangle \times \langle \mathbf{1}| \times |\mathbf{0}\rangle + \frac{1}{\sqrt{2}} \cdot |\mathbf{1}\rangle \times \langle \mathbf{0}| \times |\mathbf{0}\rangle + (-\frac{1}{\sqrt{2}}) \cdot |\mathbf{1}\rangle \times \langle \mathbf{1}| \times |\mathbf{0}\rangle$$
(by replacements of $\boldsymbol{B}_0, \boldsymbol{B}_1, \boldsymbol{B}_2$, and $\boldsymbol{B}_3$)

$$= \frac{1}{\sqrt{2}} \cdot |\mathbf{0}\rangle + \frac{1}{\sqrt{2}} \cdot |\mathbf{1}\rangle \quad \text{(by law L1)}$$

$$(H \otimes I) \times (|\mathbf{0}\rangle \otimes |\mathbf{0}\rangle)$$

$$= (H \times |\mathbf{0}\rangle) \otimes (I \times |\mathbf{0}\rangle) \quad \text{(by law L16)}$$

$$= (\frac{1}{\sqrt{2}} \cdot |\mathbf{0}\rangle + \frac{1}{\sqrt{2}} \cdot |\mathbf{1}\rangle) \otimes |\mathbf{0}\rangle \quad \text{(by the result of } H \times |\mathbf{0}\rangle \text{ and law L11)}$$

$$= \frac{1}{\sqrt{2}} \cdot |\mathbf{0}\rangle \otimes |\mathbf{0}\rangle + \frac{1}{\sqrt{2}} \cdot |\mathbf{1}\rangle \otimes |\mathbf{0}\rangle \quad \text{(by law L18)}$$

$$CX \times ((H \otimes I) \times (|\mathbf{0}\rangle \otimes |\mathbf{0}\rangle))$$

$$= (B_0 \otimes I + B_3 \otimes X) \times (\frac{1}{\sqrt{2}} \cdot |\mathbf{0}\rangle \otimes |\mathbf{0}\rangle + \frac{1}{\sqrt{2}} \cdot |\mathbf{1}\rangle \otimes |\mathbf{0}\rangle)$$
(by replacement of $CX$, and the result of $(H \otimes I) \times (|\mathbf{0}\rangle \otimes |\mathbf{0}\rangle)$)

$$= (B_0 \otimes I) \times (\frac{1}{\sqrt{2}} \cdot |\mathbf{0}\rangle \otimes |\mathbf{0}\rangle) + (B_0 \otimes I) \times (\frac{1}{\sqrt{2}} \cdot |\mathbf{1}\rangle \otimes |\mathbf{0}\rangle) + (B_3 \otimes X) \times (\frac{1}{\sqrt{2}} \cdot |\mathbf{0}\rangle \otimes |\mathbf{0}\rangle) +$$

$$B_3 \otimes X \times (\frac{1}{\sqrt{2}} \cdot |\mathbf{1}\rangle \otimes |\mathbf{0}\rangle) \quad \text{(by laws L14 and L15)}$$

$$= \frac{1}{\sqrt{2}} \cdot (B_0 \times |\mathbf{0}\rangle) \otimes (I \times |\mathbf{0}\rangle) + \frac{1}{\sqrt{2}} \cdot (B_0 \times |\mathbf{1}\rangle) \otimes (I \times |\mathbf{0}\rangle) + \frac{1}{\sqrt{2}} \cdot (B_3 \times |\mathbf{0}\rangle) \otimes (X \times |\mathbf{0}\rangle) +$$

$$\frac{1}{\sqrt{2}} \cdot (B_3 \times |\mathbf{1}\rangle) \otimes (X \times |\mathbf{0}\rangle) \quad \text{(by laws L8, L9, and L16)}$$

$$= \frac{1}{\sqrt{2}} \cdot |\mathbf{0}\rangle \otimes |\mathbf{0}\rangle + \frac{1}{\sqrt{2}} \cdot |\mathbf{1}\rangle \otimes |\mathbf{1}\rangle \quad \text{(by replacements of } B_0, B_3, \text{ and } X, \text{ and laws L1, L11,}$$
and L15)

Using the laws, the term is reduced to a normal form that is a linear combination of the tensor product of the standard basis with scalars. The whole process is conducted automatically in Maude and the result is the same as expected. The key idea is to reduce the matrix multiplication in the form of $\langle i|j\rangle$ into a scalar and simplify the matrix representation by absorbing ones and eliminating zeros (see law L3). In this manner, our symbolic reasoning about matrices can be conducted automatically by rewriting in Maude instead of explicitly calculating matrices.

## FORMAL SPECIFICATION

This section shows how we specify in Maude qubits, quantum gates, measurements, and then quantum circuits in order to symbolically model check quantum circuits in a generic way.

## Maude specification of qubits, gates, and measurements

Qubits are specified in Maude as the linear combination of tensor product of the standard basis in Dirac notation with scalars and similarly for quantum gates. Because $|0\rangle$ and $|1\rangle$ can be viewed as $2 \times 1$ matrices, then qubits and quantum gates are basically matrices. Quantum gates act on qubits (a quantum state) specified in Maude as a matrix multiplication with a deterministic transition in Maude. In this article, we only consider binary projective measurements on the standard basis, and thus the measurement operators are $\{M_0, M_1\}$. A measurement of a single qubit in a quantum state is specified in Maude by two state transitions with probabilities $p(m)$ for $m \in \{0, 1\}$, making a non-deterministic probabilistic transition. Each of the two transitions shows how its measurement operator acts on the single qubit in a state and is specified similarly as quantum gates, however, with respect to the probabilities.

## A generic maude specification of quantum circuits

Quantum circuits are composed of a sequence of quantum gates, measurements, initializations of qubits, and possibly other actions. In this article, we consider the specification of the whole quantum state of a quantum circuit, the classical bits obtained from measurements, and the sequence of quantum gates, measurements, and conditional gates describing how a quantum circuit works. We then build Kripke structures for quantum circuits in order to conduct model checking that quantum circuits satisfy desired properties. Some essential elements are shared in the Kripke structures, making a first step toward a general framework for specifying and verifying quantum circuits.

### *Elements of quantum circuits*

*A whole quantum state* of a quantum circuit is specified in Maude as a collection of qubits associated with indices in circuits, where each element is one of the forms as follows:

- `(q[`$i$`]:` $\psi\langle.\rangle)|$ denotes a single qubit in state $|\psi\rangle$ at $q_i$,
- `(q[`$i,...,j$`]:` $\psi\langle.\rangle)|$ denotes a single qubit in state $|\psi\rangle$ at $q_i$, denotes an entangled state in state $|\psi\rangle$ at $q_i, ..., q_j$, where the order of $i, ..., j$ is relevant.

Note that $q_i$ and $q_j$ denote the labels of quantum wires (refer to our circuits in 'Symbolic Model Checking' for more visualization), where $i$ and $j$ represent the indices of the qubits in the whole quantum state of a quantum circuit.

*Classical bits* are specified in Maude as a map from indices in circuits to Boolean values, where each entry is in the form of `(`$i \mapsto b$`)`, meaning that the value of the classical bit stored at $c_i$ is $b$ whose value is either 0 or 1.

*A sequence of quantum gates, measurements*, and *conditional gates* in a quantum circuit is specified in Maude as a list of actions in which each action is one of the forms as follows:

- `I(`$i$`)` applies the $I$ gate on $q_i$,
- `X(`$i$`)` applies the $X$ gate on $q_i$,
- `Y(`$i$`)` applies the $Y$ gate on $q_i$,
- `Z(`$i$`)` applies the $Z$ gate on $q_i$,
- `H(`$i$`)` applies the $H$ gate on $q_i$,

- CX($i,j$) applies the **CX** gate on $q_i$ and $q_j$,
- M($i$) measures $q_i$ with the standard basis,
- c[$i$] == $b$ ? AL checks if the classical bit at $c_i$ equals $b$, then a list AL of actions is executed.

Although our specification supports some additional gates, including *S*, *T*, *CY*, *CZ*, *SWAP*, *CCY*, *CCZ*, and *CSWAP* gates, we do not mention them here because the additional gates are not used for our case studies in this article. Note that those gates can form universal quantum gates, meaning that we may use those gates to describe universal quantum computation. However, we need to enhance our symbolic reasoning for complex numbers because its specification is not complete in this article. Based on the actions specified above, we can describe the circuits for several quantum communication protocols as shown in 'Symbolic Model Checking'. The reader who is interested in how quantum computation works with our specification can refer to Appendix B for more details.

### Kripke structures of quantum circuits

Let *K* be the Kripke structure specifying a quantum circuit. There are five kinds of observable components in our specification as follows:

- (qstate: $qs$) represents the whole quantum state $qs$,
- (bits: $bm$) indicates the classical bits obtained from measurements and stored in a bit map $bm$,
- (prob: $p$) denotes the probability $p$ at the current quantum state,
- (actions: $al$) signifies the action list $al$, guiding us on how the circuit works,
- (isEnd: $b$) designates termination with Boolean flag $b$.

Each state in *S* is expressed as {*obs*}, where *obs* is a collection of those observable components consisting of one qstate observable component, one prob observable component, one bits observable component, one actions observable component, and one isEnd observable component. Note that the whole quantum state denotes the quantum state of a quantum circuit, while each state in *S* denotes a state under model checking, which consists of not only the whole quantum state but also other necessary information for model checking.

The set *T* of transitions is specified in Maude by eleven rewrite rules in our specification. Let OCs be a Maude variable of observable component collections, Q and Q' be Maude variables of whole quantum states, BM be a Maude variable of bit maps, Prob and Prob' be Maude variables of scalars, AL and AL' be Maude variables of action lists, B be a Maude variable of Boolean values, and N, N1, and N2 are Maude variables of natural numbers.

The first six rewrite rules are as follows:

```
rl [I] : {(qstate: Q) (actions: (I(N) AL)) OCs}
=> {(qstate: Q) (actions: AL) OCs} .

crl [X] : {(qstate: Q) (actions: (X(N) AL)) OCs}
=> {(qstate: Q') (actions: AL) OCs}
if Q' := (Q).X(N) .

crl [Y] : {(qstate: Q) (actions: (Y(N) AL)) OCs}
=> {(qstate: Q') (actions: AL) OCs}
if Q' := (Q).Y(N) .
```

```
crl [Z] : {(qstate: Q) (actions: (Z(N) AL)) OCs}
=> {(qstate: Q') (actions: AL) OCs}
if Q' := (Q).Z(N) .

crl [H] : {(qstate: Q) (actions: (H(N) AL)) OCs}
=> {(qstate: Q') (actions: AL) OCs}
if Q' := (Q).H(N) .

crl [CX] : {(qstate: Q) (actions: (CX(N1, N2) AL)) OCs}
=> {(qstate: Q') (actions: AL) OCs}
if Q' := (Q).CX(N1, N2) .
```

The rules `I` , `X` , `Y` , `Z` , `H`, and `CX`  simulate how the $I, X, Y, Z, H$, and $CX$ gates act on the whole quantum state in the `qstate`  observable component if its action appears in the `actions`  observable component, respectively. In this specification, we consider the probabilities of measurements in quantum computation and so we can analyze not only qualitative properties but also quantitative properties for quantum circuits.

The next two rewrite rules are as follows:

```
crl [M0] : {(qstate: Q) (actions: (M(N) AL)) (prob: Prob) (bits: BM) OCs}
=> {(qstate: Q') (actions: AL) (prob: (Prob .* Prob'))
    (bits: insert(N, 0, BM)) OCs}
if {qstate: Q', prob: Prob'} := (Q).M(P0,N) .

crl [M1] : {(qstate: Q) (actions: (M(N) AL)) (prob: Prob) (bits: BM) OCs}
=> {(qstate: Q') (actions: AL) (prob: (Prob .* Prob'))
    (bits: insert(N, 1, BM)) OCs}
if {qstate: Q', prob: Prob'} := (Q).M(P1,N) .
```

The rules `M0`  and `M1`  say that we measure the qubit at index `N`  with the measurement operators $M_0$ and $M_1$, respectively; the classical outcomes are stored accordingly into the bit map in the `bits`  observable component; the probabilities and the post-measurement states are also updated in the `prob`  and `qstate`  observable components, respectively. These two rules make a non-deterministic probabilistic transition when measuring a single qubit.

The next rewrite rule describes how to conditionally perform the next actions based on classical bits obtained from measurements if applicable.

```
rl [cif]:
{(qstate: Q) (bits: ((N |-> N1),BM)) (actions: ((c[N] == N2 ? AL') AL)) OCs}
=> {(qstate: Q) (bits: ((N |-> N1), BM))
    (actions: ((if (N1 == N2) then AL' else nil fi) AL)) OCs} .
```

This rule says that if `c[N] x== N2 ? AL'`  is in the action list and the classical bit `N1`  at index `N`  equals the conditional value `N2` , then the action list `AL'`  is prepended to the action list `AL`  in the `actions`  observable component to be executed next; otherwise, it is ignored.

The last two rules are as follows:

```
rl [end]: {(actions: nil) (isEnd: false) OCs}
=> {(actions: nil) (isEnd: true) OCs} .

rl [stutter]: {(isEnd: true) OCs}
=> {(isEnd: true) OCs} .
```

The rule `end`  marks the termination if the action list is `nil` , meaning no more action. Meanwhile, the rule `stutter`  is necessary to make $T$ total when the `isEnd`  observable component is true.

For Kripke structure $K = \langle S, I, T, A, L \rangle$ of a quantum circuit, we can reuse $S$ and $T$, while $I$ is required to define initial states, and $A$ and $L$ are required to define desired properties

for the quantum circuit. Therefore, our specification can be a first step toward a general framework to formally specify and verify quantum circuits in Maude.

# SYMBOLIC MODEL CHECKING

We have used our symbolic approach to conduct model checking for several quantum communication protocols in the early stage of quantum communication:

- Superdense Coding introduced by *Bennett & Wiesner (1992)* for transmitting two classical bits using an entangled state,
- Quantum Teleportation presented by *Bennett et al. (1993)* for teleporting an arbitrary pure state by sending two bits of classical information,
- Quantum Secret Sharing developed by *Hillery, Bužek & Berthiaume (1999)* for teleporting a pure state from a sender (Alice) to a receiver (Bob) with the help of a third party (Charlie),
- Entanglement Swapping proposed by *Zukowski et al. (1993)* for creating a new entangled state,
- Quantum Gate Teleportation suggested by *Gottesman & Chuang (1999)* for teleporting two arbitrary states through the controlled-NOT gate,
- Two Mirror-image Teleportation devised by *Williams (2008)* for teleporting two arbitrary states,
- Quantum Network Coding originated by *Satoh, Gall & Imai (2012)* for sending two entangled states simultaneously.

Superdense Coding is the simplest one that uses only two qubits; Quantum Teleportation uses three qubits; Quantum Secret Sharing proposed relying on the mechanism of Quantum Teleportation uses four qubits; Entanglement Swapping uses four qubits; Quantum Gate Teleportation uses six qubits; Two Mirror-image Teleportation uses six qubits; and Quantum Network Coding uses ten qubits.

For the sake of simplicity, this section demonstrates how to use our symbolic approach to conduct model checking experiments for four quantum communication protocols: Superdense Coding, Quantum Teleportation, Quantum Secret Sharing, and Quantum Gate Teleportation. Meanwhile, other communication protocols are similar and the full specifications of all quantum communication protocols concerned in this article are publicly available at https://doi.org/10.5281/zenodo.10783951. For each case study, we only need to specify $I$, $A$, and $L$ to model check that $K$ satisfies desired properties, while $S$ and $T$ in $K$ are reused as described in the previous section. In this section, we use `qstate(.)` and `qubitAt(.)` as two functions to get the whole quantum state from a state in $S$ and to get a single qubit at some index from the whole quantum state, respectively, where the symbol . denotes its parameter. It is important to note that we use quantum circuits to represent the quantum communication protocols. Therefore, sending or receiving classical bits obtained from the measurement outcomes will be abstracted away. However, in the following introduction of each protocol, we describe how the protocol works, assuming that participants can communicate with each other (*e.g.*, Alice can send a classical bit to Bob). This makes it easier for the reader to understand how each protocol works.

## Superdense coding

### *Introduction*

Superdense Coding (SC) introduced by *Bennett & Wiesner (1992)* takes advantage of entanglement in quantum mechanics to send two classical bits from Alice to Bob using just a pair of entangled qubits. Figure 2 depicts the circuit for Superdense Coding. The single wires denote qubits referred to as $q_i$, while the double wires denote classical bits referred to as $c_i$. Alice acts on $q_0$ while Bob acts on $q_1$ as follows:

- First, $q_0$ are $q_1$ are initially in the basic state $|0\rangle$. We need to prepare an entangled state between $q_0$ and $q_1$ by applying the sequence of the **H** gate on $q_0$ and the **CX** gate on $q_0$ and $q_1$. The entangled state is shared between Alice and Bob using a quantum channel, where $q_0$ and $q_1$ are manipulated by Alice and Bob, respectively.
- Second, Alice needs to send two classical bits $x$ and $y$, where $x, y \in \{0, 1\}$, as depicted in Fig. 2. Depending on the values of $x$ and $y$ that Alice wants to send to Bob, Alice will apply the $\sigma_i$ gate on $q_0$, where $i = y + x * (2 + (-1)^y)$ ranging over $\{0, 1, 2, 3\}$ and $\sigma_0, \sigma_1, \sigma_2$, and $\sigma_3$ are $I, X, Y, Z$ gates, respectively.
- Third, we then apply the sequence of the **CX** gate on $q_0$ and $q_1$, and the **H** gate on $q_0$.
- Fourth, we measure the qubits $q_0$ and $q_1$, and immediately obtain two classical outcomes (0 or 1) stored in $c_0$ and $c_1$, respectively.

At the end, the pair $(c_0, c_1)$ of classical bits obtained from Bob is expected to be the same as the pair $(x, y)$ of classical bits sent by Alice. We would like to verify the correctness of Superdense Coding by using our symbolic model checking.

### *Specification of superdense coding*

Regarding the actions specified in 'Formal Specification', we can describe the circuit for Superdense Coding with different values of classical bits used for $(x, y)$ as follows:

- $(x, y) = (0, 0)$ with $\sigma_0 = I$:
  H(0) CX(0, 1) I(0) CX(0, 1) H(0) M(0) M(1)
- $(x, y) = (0, 1)$ with $\sigma_1 = X$:
  H(0) CX(0, 1) X(0) CX(0, 1) H(0) M(0) M(1)
- $(x, y) = (1, 1)$ with $\sigma_2 = Y$:
  H(0) CX(0, 1) Y(0) CX(0, 1) H(0) M(0) M(1)
- $(x, y) = (1, 0)$ with $\sigma_3 = Z$:
  H(0) CX(0, 1) Z(0) CX(0, 1) H(0) M(0) M(1)

Let $I_{SC}$ be the set of initial states for Superdense Coding. It consists of four initial states corresponding to the four possible values used for $(x, y)$ as follows:

```
{(isEnd: false)
(prob: 1)
(qstate: (q[0]: |0>) (q[1]: |0>))
(bits: empty)
(actions: H(0) CX(0, 1) I(0) CX(0, 1) H(0)
          M(0) M(1))}

{(isEnd: false)
(prob: 1)
(qstate: (q[0]: |0>) (q[1]: |0>))
(bits: empty)
```

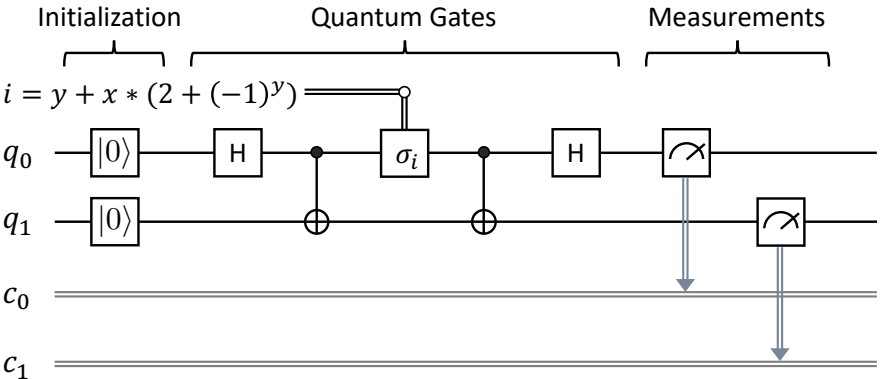

**Figure 2  Superdense coding.**

```
(actions: H(0) CX(0, 1) X(0) CX(0, 1) H(0)
          M(0) M(1))}

{(isEnd: false)
(prob: 1)
(qstate: (q[0]: |0>) (q[1]: |0>))
(bits: empty)
(actions: H(0) CX(0, 1) Y(0) CX(0, 1) H(0)
          M(0) M(1))}

{(isEnd: false)
(prob: 1)
(qstate: (q[0]: |0>) (q[1]: |0>))
(bits: empty)
(actions: H(0) CX(0, 1) Z(0) CX(0, 1) H(0)
          M(0) M(1))}
```

Let us refer to the four initial states as `init0`, `init1`, `init2`, and `init3`, respectively. Initially, for each initial state, the `isEnd` observable component is false, the `prob` observable component is one, the `qstate` is the basic state (saying $|00\rangle$), while the `actions` observable component contains the action list describing how Superdense Coding works with respect to the values of classical bits $x$ and $y$.

### Model checking superdense coding

Let $K_{SC}$ be the Kripke structure for Superdense Coding. To model check that $K_{SC}$ satisfies desired properties, we specify $A_{SC}$ and $L_{SC}$ for Superdense Coding. $A_{SC}$ has four atomic propositions `isGateI`, `isGateX`, `isGateY`, and `isGateZ`. $L_{SC}$ is specified as follows:

```
eq {(isEnd: true) (bits: BM) (prob: Prob) OCs} |= isGateI
= (Prob > 0) implies (BM[0] == 0 and BM[1] == 0) .

eq {(isEnd: true) (bits: BM) (prob: Prob) OCs} |= isGateX
= (Prob > 0) implies (BM[0] == 0 and BM[1] == 1) .

eq {(isEnd: true) (bits: BM) (prob: Prob) OCs} |= isGateY
= (Prob > 0) implies (BM[0] == 1 and BM[1] == 1) .

eq {(isEnd: true) (bits: BM) (prob: Prob) OCs} |= isGateZ
= (Prob > 0) implies (BM[0] == 1 and BM[1] == 0) .

eq {OCs} |= PROP = false [owise] .
```

where `BM` and `Prob` are Maude variables denoting the classical bit map and the probability at a state in $S$, respectively.

The five equations say that `isGateI` holds at a state if the state contains `(isEnd: true)`, `(bits: BM)`, and `(prob: Prob)` such that the condition `BM[0] == 0 and BM[1] == 0` holds whenever `Prob > 0` (a non-zero probability), meaning that the pair $(0, 0)$ of classical bits obtained from Bob is the same as the classical bits sent by Alice when the gate **X** is used; and similar for other propositions. Let `gateIProp`, `gateXProp`, `gateYProp`, and `gateZProp` be LTL formulas defined as `<> isGateI`, `<> isGateX`, `<> isGateY`, and `<> isGateZ`, respectively, where `<>` is the eventual temporal connective.

We model check that $K_{SC} = \langle S, I_{SC}, T, A_{SC}, L_{SC} \rangle$ satisfies `gateIProp`, `gateXProp`, `gateYProp`, and `gateZProp` from the initial states `init0`, `init1`, `init2`, and `init3`, respectively, in Maude as follows:

```
red modelCheck(init0, gateIProp) .
red modelCheck(init1, gateXProp) .
red modelCheck(init2, gateYProp) .
red modelCheck(init3, gateZProp) .
```

No counterexample is found in just 1 ms for each model checking experiment and so $K_{SC}$ satisfies `gateIProp`, `gateXProp`, `gateYProp`, and `gateZProp`. In other words, for all possible values of $(x, y)$ and its corresponding gates used, Bob can receive the same classical values sent by Alice at the end. Thus, we successfully verify the correctness of Superdense Coding by using our symbolic model checking approach. Note that we do not treat the input $(x, y)$ to Superdense Coding as a two-bit value symbolically; instead, we conduct four separate model checking experiments. This is necessary because we need to instantiate the input to determine which quantum circuit and its desired property should be considered individually.

Moreover, we conduct some more model checking experiments for Superdense Coding to confirm that Bob cannot receive bits that differ from the ones sent by Alice using the following commands:

```
red modelCheck(init0, gateXProp) .
red modelCheck(init0, gateYProp) .
red modelCheck(init0, gateZProp) .
red modelCheck(init1, gateIProp) .
red modelCheck(init1, gateYProp) .
red modelCheck(init1, gateZProp) .
red modelCheck(init2, gateIProp) .
red modelCheck(init2, gateXProp) .
red modelCheck(init2, gateZProp) .
red modelCheck(init3, gateIProp) .
red modelCheck(init3, gateXProp) .
red modelCheck(init3, gateYProp) .
```

where each of `init0`, `init1`, `init2`, and `init3` is used to check with other properties compared to the previous experiments. Each command returns a counterexample that confirms that Bob cannot receive bits that differ from the ones sent by Alice.

## Quantum teleportation
### Introduction
Quantum Teleportation (QT) introduced by *Bennett et al. (1993)* also takes advantage of entanglement in quantum mechanics to send an unknown quantum state $|\psi\rangle$ from Alice

to Bob by using only three qubits and two classical bits. Because the no-cloning theorem, as stated in *Wootters & Zurek (1982)*, does not allow copying an arbitrary unknown quantum state, the protocol becomes extremely important to transmit an arbitrary unknown quantum state from one source to another. The difference between Superdense Coding and Quantum Teleportation is that the former transmits two classical bits, while the latter transmits an arbitrary unknown quantum state.

The circuit depicted in Fig. 3 shows how the protocol works. Alice acts on $q_0$ and $q_1$, and Bob acts on $q_2$ as follows:

- First, we prepare an unknown state $|\psi\rangle = \alpha|0\rangle + \beta|1\rangle$ at $q_0$, where $\alpha$ and $\beta$ are complex numbers such that $|\alpha|^2 + |\beta|^2 = 1$. Initially, $q_1$ and $q_2$ are in the state $|0\rangle$.
- Second, we apply a sequence of quantum gates to manipulate three qubits. In this case, we only consider the single-qubit Hadamard **H** and two-qubit controlled-NOT **CX** gates. We first apply the **H** gate on $q_1$ followed by the **CX** gate on $q_1$ and $q_2$ in order to make an entangled state shared between Alice and Bob. Alice then applies the **CX** gate on $q_0$ and $q_1$ followed by the **H** gate on $q_0$.
- Third, we measure the qubits $q_0$ and $q_1$, and immediately obtain two classical outcomes (0 or 1) stored in $c_0$ and $c_1$, respectively.
- Fourth, we conditionally apply single-qubit **X** and **Z** gates on $q_2$ depending on the two classical bits in $c_0$ and $c_1$. Concretely, we use the **X** gate if $c_1$ equals one and followed by the **Z** gate if $c_0$ equals one.

At the end, Bob will have $|\psi\rangle$ and Alice will not have it anymore. We would like to verify whether Alice can correctly send an arbitrary unknown quantum state to Bob at the end by using our symbolic model checking.

***Specification of quantum teleportation***

We can describe the circuit for Quantum Teleportation based on the actions specified in 'Formal Specification' as follows:

```
H(1) CX(1, 2) CX(0, 1) H(0) M(0) M(1) (c[1] == 1 ? X(2)) (c[0] == 1 ? Z(2))
```

Let $I_{QT}$ be the set of initial states for Quantum Teleportation. It consists of only one initial state as follows:

```
{(isEnd: false)
(prob: 1)
(qstate: (q[0]: a . |0> + b . |1>)
         (q[1]: |0>) (q[2]: |0>))
(bits: empty)
(actions: H(1) CX(1, 2) CX(0, 1) H(0)
          M(0) M(1)
          c[1] == 1 ? X(2)
          c[0] == 1 ? Z(2))}
```

where `a` and `b` are Maude constants denoting arbitrary scalars such that $|a|^2 + |b|^2 = 1$. Initially, the `isEnd` observable component is false, the `prob` observable component is one, the `qstate` is a symbolic state that is the same as the input state of the protocol, the `actions` observable component contains the action list describing how the protocol works.

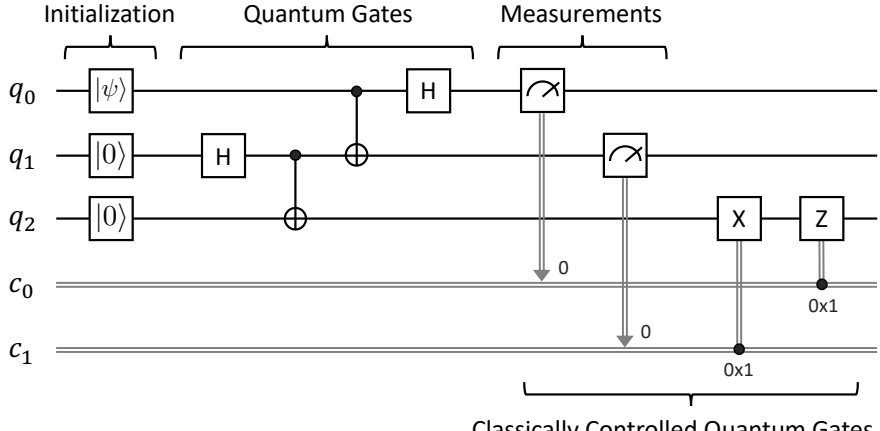

**Figure 3** **Quantum teleportation.**

### Model checking quantum teleportation

Let $K_{QT}$ and init be the Kripke structure and the initial state for Quantum Teleportation, respectively. To model check that $K_{QT}$ satisfies desired properties, we specify $A_{QT}$ and $L_{QT}$. $A_{QT}$ has one atomic proposition isSuccess. $L_{QT}$ is specified as follows:

```
eq {(isEnd: true) (qstate: Q) (prob: Prob) OCs} |= isSuccess
= Prob > 0 implies (qubitAt(Q, 2) == qubitAt(qstate(init), 0) and
  qubitAt(Q, 0) =/= qubitAt(qstate(init), 0)) .
eq {OCs} |= PROP = false [owise] .
```

where Q and Prob are Maude variables denoting the whole quantum state and the probability at a state, respectively.

The two equations say that isSuccess holds at a state if the state contains (isEnd: true), (qstate: Q), and (prob: Prob) such that the condition qubitAt(Q, 2) == qubitAt(qstate(init), 0) and qubitAt(Q, 0) == qubitAt(qstate(init), 0) holds whenever Prob > 0 holds, meaning that at the end, the qubit received by Bob is equal to the qubit sent by Alice at the beginning, and Alice does not have the qubit anymore with a non-zero probability. Let teleProp be an LTL formula defined as <> isSuccess.

We want to model check that $K_{QT} = \langle S, I_{QT}, T, A_{QT}, L_{QT} \rangle$ satisfies teleProp from the initial state init in Maude as follows:

```
red modelCheck(init, teleProp) .
```

No counterexample is found in just 3 ms and so $K_{QT}$ satisfies teleProp. In other words, we successfully verify the correctness of Quantum Teleportation by using our symbolic model checking approach.

During LTL model checking, each state in a computation reachable from an initial state contains information on the probability accumulated up to the state from previous states. Ultimately, we obtain the probability of each computation. We would like to additionally check that if a computation has a probability greater than 0, the probability is also less

than 1/2. We achieve this by adding the condition `Prob > 1/2` to the condition of the first equation specifying the labeling function for the atomic proposition `isSuccess`.

The first equation now becomes as follows:

```
eq {(isEnd: true) (qstate: Q) (prob: Prob) OCs} |= isSuccess
= Prob > 0 implies (qubitAt(Q, 2) == qubitAt(qstate(init), 0) and
  qubitAt(Q, 0) =/= qubitAt(qstate(init), 0) and Prob < 1/2) .
```

We then conduct the model checking experiment again and no counterexample is found because each computation indeed has the probability of 1/4 equally. This artificial model checking experiment demonstrates that we can specify quantitative properties by considering the accumulated probability across states for each computation.

Moreover, we conduct one more model checking experiment to confirm that Alice indeed keeps her initial qubit with zero probability by changing the condition `qubitAt(Q, 0) =/= qubitAt(qstate(init), 0)` to `qubitAt(Q, 0) == qubitAt(qstate(init), 0)` in the condition of the first equation specifying the labeling function for the atomic proposition `isSuccess`, and conducting the model checking experiment again. As expected, a counterexample was found, showing that Alice indeed keeps her initial qubit with zero probability.

### Reachability analysis for quantum teleportation

Besides using LTL model checking, we can also use reachability analysis to verify the correctness of Quantum Teleportation with the same property mentioned above. Maude is equipped with the search command with which reachability analysis can be conducted. The following search command, where `TELEPORT` is the specification of Quantum Teleportation, `init` is the initial state for `TELEPORT`,

```
search in TELEPORT: init
=>* {(qstate: Q) (isEnd: true) (prob: P) OCs}
such that not (
  Prob > 0 implies (qubitAt(Q, 2) == qubitAt(qstate(init), 0) and
  qubitAt(Q, 0) =/= qubitAt(qstate(init), 0))
) .
```

finds all states reachable from the initial state that contain `(isEnd: true)`, `(qstate: Q)`, and `(prob: Prob)` such that it is not the case where the qubit received by Bob at the end is equal to the qubit sent by Alice at the beginning, and Alice does not have the qubit anymore with a non-zero probability. Note that the condition used here is the negation of the condition used to define the atomic proposition `isSuccess` above. The search command does not find any state in just 2 ms, meaning that we successfully verify the correctness of Quantum Teleportation by using reachability analysis.

The reason why we use the Maude LTL model checker is because it is convenient to express desired properties of quantum circuits in LTL. Quantum circuits look simple but have non-determinism because of measurements, and then have multiple possible execution paths. It is necessary to take all such multiple execution paths. Desired properties of quantum circuits are in the form: for each possible execution path, something good eventually happens. For example, a qubit at the final state for each possible execution path is the same as another qubit at the initial state with a non-zero probability. Meanwhile, we

need to find all states that do not satisfy the condition and we need to think in an inverse way when the search command is used. As written, we can use the search command, but because times taken by the search command (*i.e.,* 2 ms) and the Maude LTL model checker (*i.e.,* 3 ms) are comparable and it is convenient to express desired properties of quantum circuits in LTL, so we use the Maude LTL model checker.

## Quantum secret sharing

### *Introduction*

Quantum Secret Sharing (QSS) was first invented by *Hillery, Bužek & Berthiaume (1999)* and some attempts to describe quantum circuits for the protocol were presented by *Joy et al. (2019)*. This protocol also takes advantage of entanglement to send an unknown quantum state $|\psi\rangle$ from Alice to either Bob or Charlie using the mechanism of Quantum Teleportation. However, this protocol uses four qubits and three classical bits, and especially neither Bob nor Charlie can independently reconstruct $|\psi\rangle$ by themselves. The one wants to retrieve the unknown quantum state $|\psi\rangle$ if and only if some information from the other is provided. This protocol has been used in many applications, such as quantum money schemes introduced by *Wiesner (1983)*, *Wang et al. (2007)*, quantum error-correcting codes introduced by *Cleve, Gottesman & Lo (1999)*, *Matsumoto (2017)*, and a graph-theoretic protocol introduced by *Sarvepalli (2012)*, *Gravier et al. (2015)*, demonstrating its importance.

The circuit for Quantum Secret Sharing is depicted in Fig. 4. We suppose that Charlie will reconstruct the unknown quantum state $|\psi\rangle$ sent by Alice with consent from Bob in the following description. Alice acts on $q_0$ and $q_1$, Bob acts on $q_2$, and Charlie acts on $q_3$ as follows:

- First, we prepare an arbitrary unknown state $|\psi\rangle = \alpha|0\rangle + \beta|1\rangle$ at $q_0$, where $\alpha$ and $\beta$ are complex numbers such that $|\alpha|^2 + |\beta|^2 = 1$. Initially, $q_1$, $q_2$, and $q_3$ are in the basic state $|0\rangle$.
- Second, we apply a sequence of quantum gates to manipulate four qubits. We first apply the *H* gate on $q_1$, the *CX* gate on $q_1$ and $q_2$, and the *CX* gate on $q_1$ and $q_3$ in order to make an entangled state shared between Alice, Bob, and Charlie. Alice then applies the *CX* gate on $q_0$ and $q_1$ followed by the *H* gate on $q_0$. Bob then applies the *H* gate on $q_2$ to make it possible to measure in *X-basis* (or the *diagonal* basis $\{|+\rangle, |-\rangle\}$) subsequently.
- Third, we measure the qubits $q_0$, $q_1$, and $q_2$ and immediately obtain three classical outcomes (0 or 1) stored in $c_0$, $c_1$, and $c_2$, respectively.
- Fourth, we conditionally apply single-qubit *X*, *Z*, and *Z* gates on $q_3$ depending on the three classical bits in $c_1$, $c_0$, and $c_2$. Concretely, we use the *X* gate if $c_1$ equals one and similarly for others. We can see that Charlie also needs to use the measurement outcome from Bob in order to reconstruct $|\psi\rangle$ in this step.

At the end, Charlie will have $|\psi\rangle$ with consent from Bob, and Alice will not have it anymore. We would like to verify whether Charlie can correctly reconstruct an arbitrary unknown quantum state sent by Alice with consent from Bob at the end by using our symbolic model checking. Note that this property is one aspect of QSS because we do

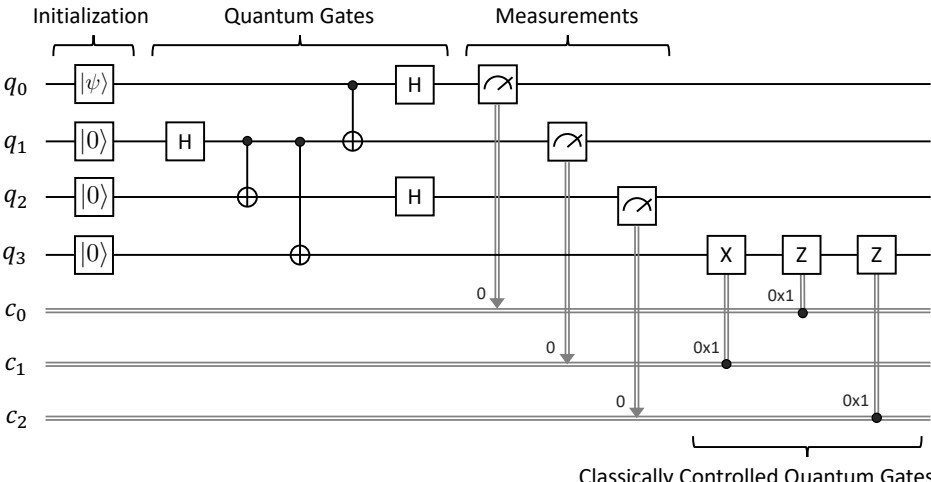

**Figure 4** **Quantum secret sharing.**

not consider the following property under verification: neither Bob nor Charlie can independently reconstruct $|\psi\rangle$ by themselves.

### Specification of quantum secret sharing

We can describe the circuit for Quantum Secret Sharing based on the actions specified in 'Formal Specification' as follows:

```
H(1) CX(1, 2) CX(1, 3) CX(0, 1) H(0) H(2) M(0) M(1) M(2)
(c[1] == 1 ? X(3)) (c[0] == 1 ? Z(3)) (c[2] == 1 ? Z(3))
```

Let $I_{QSS}$ be the set of initial states for Quantum Secret Sharing. It consists of only one initial state as follows:

```
{(isEnd: false)
(prob: 1)
(qstate: (q[0]: a . |0> + b . |1>)
         (q[1]: |0>) (q[2]: |0>) (q[3]: |0>))
(bits: empty)
(actions:
   H(1) CX(1, 2) CX(1, 3) CX(0, 1) H(0) H(2)
   M(0) M(1) M(2)
   c[1] == 1 ? X(3)
   c[0] == 1 ? Z(3)
   c[2] == 1 ? Z(3))}
```

where a and b are Maude constants denoting arbitrary scalars such that $|a|^2 + |b|^2 = 1$. Initially, the `isEnd` observable component is false, the `prob` observable component is one, the `qstate` is a symbolic state that is the same as the input state of the protocol, the `actions` observable component contains the action list describing how the protocol works.

### Model checking quantum secret sharing

Let $K_{QSS}$ and `init` be the Kripke structure and the initial state for Quantum Secret Sharing, respectively. To model check that $K_{QSS}$ satisfies desired properties, we specify $A_{QSS}$ and $L_{QSS}$. $A_{QSS}$ has one atomic proposition `isSuccess`. $L_{QSS}$ is specified as follows:

```
eq {(isEnd: true) (qstate: Q) (prob: Prob) OCs} |= isSuccess
= Prob > 0 implies (qubitAt(Q, 3) == qubitAt(qstate(init), 0) or
                   (-1) . qubitAt(Q, 3) == qubitAt(qstate(init), 0)).
eq {OCs} |= PROP = false [owise] .
```

where `Q` and `Prob` are Maude variables denoting the whole quantum state and the probability at a state, respectively.

The two equations say that `isSuccess` holds at a state if the state contains `(isEnd: true)`, `(qstate: Q)`, and `(prob: Prob)` such that the condition `qubitAt(Q, 3) == qubitAt(qstate(init), 0)` or `qubitAt(Q, 3) == qubitAt(qstate(init), 0)` holds whenever `Prob > 0` holds, meaning that the qubit received by Charlie with consent from Bob at the end is equal to the qubit sent by Alice at the beginning with a non-zero probability. Note that a factor $\gamma$ on a quantum state for which $|\gamma| = 1$ is regarded as a *global phase* and quantum states that differ only by a global phase are physically indistinguishable and equivalent as shown in *Nielsen & Chuang (2010)*. That is why we use `(-1)` . `qubitAt(Q, 3) == qubitAt(qstate(init), 0)` in addition to `qubitAt(Q, 3) == qubitAt(qstate(init), 0)` in the condition. Let `secretProp` be an LTL formula defined as `<> isSuccess`.

We want to model check that $K_{QSS} = \langle S, I_{QSS}, T, A_{QSS}, L_{QSS} \rangle$ satisfies `secretProp` from the initial state `init` in Maude as follows:

```
red modelCheck(init, secretProp) .
```

No counterexample is found in just 13 ms and so $K_{QSS}$ satisfies `secretProp` In other words, we successfully verify the correctness of the Quantum Secret Sharing by using our symbolic model checking approach.

## Quantum gate teleportation
### Introduction
Quantum Gate Teleportation (QGT) is a generalization of quantum teleportation invented by *Gottesman & Chuang (1999)* for teleporting two arbitrary states through the controlled-NOT gate with the use of six qubits and four classical bits. This protocol can be regarded as a single technique to reduce resource requirements for quantum computers and unifies known protocols for fault-tolerant quantum computations, demonstrating its importance.

The circuit for Quantum Gate Teleportation[2] is depicted in Fig. 5. Alice acts on $q_0$ and $q_1$, Bob acts on $q_2$, $q_3$, $q_4$, and $q_5$ as follows:

- First, Alice prepares an arbitrary unknown state $|\psi\rangle = a|0\rangle + b|1\rangle$ at $q_0$, where $a$ and $b$ are complex numbers such that $|a|^2 + |b|^2 = 1$. Similarly, Bob also prepares an arbitrary unknown state $|\varphi\rangle = c|0\rangle + d|1\rangle$ at $q_5$. Initially, $q_1$, $q_2$, $q_3$, and $q_4$ are in the basic state $|0\rangle$.
- Second, we prepare an entangled state shared between Alice and Bob from $q_1$ to $q_4$ by applying a sequence of quantum gates as follows. We first apply the **H** gate on $q_1$, the **CX** gate on $q_1$ and $q_2$, the **H** gate on $q_3$, the **CX** gate on $q_3$ and $q_4$, and finally the **CX** gate on $q_3$ and $q_2$.
- Third, Alice then applies the **CX** gate on $q_1$ and $q_0$ followed by the **H** gate on $q_1$. Meanwhile, Bob applies the **CX** gate on $q_5$ and $q_4$ followed by the **H** gate on $q_5$.

[2]This circuit used here is a revised version of the original version presented in *Gottesman & Chuang (1999)*; *Ding & Chong (2020)* and the reader is recommended to refer to 'Remark on Quantum Gate Teleportation' for more details.

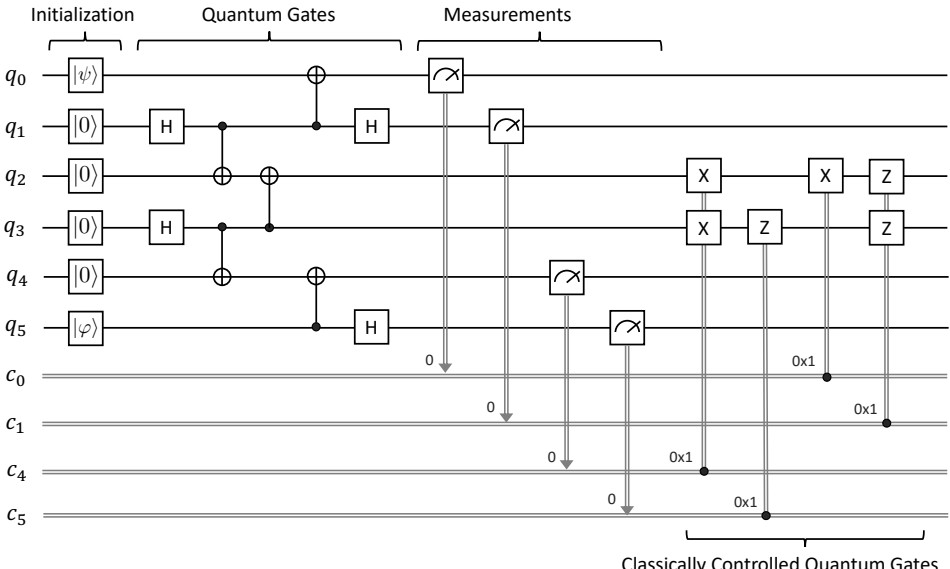

**Figure 5  Quantum gate teleportation.**

- Fourth, we measure the qubits $q_0$, $q_1$, $q_4$, and $q_5$ and immediately obtain four classical outcomes (0 or 1) stored in $c_0$, $c_1$, $c_4$, and $c_5$, respectively.
- Fifth, we conditionally apply the **X** gate on $q_2$ and $q_3$, the **Z** gate on $q_3$, the **X** gate on $q_2$, and the **Z** gate on $q_2$ and $q_3$, depending on the four classical bits in $c_4$, $c_5$, $c_0$, and $c_1$.

At the end, Bob will have the controlled-NOT gate of $|\varphi\rangle$ and $|\psi\rangle$ (*i.e.*, **CX**$(|\varphi\rangle, |\psi\rangle)$) at the indices $q_2$ and $q_3$. We would like to verify whether Alice can successfully teleport two arbitrary unknown quantum states through the controlled-NOT gate to Bob at $q_2$ and $q_3$ at the end by using our symbolic model checking.

### *Specification of quantum gate teleportation*

We can describe the circuit for Quantum Gate Teleportation based on the actions specified in 'Formal Specification' as follows:

```
H(1) CX(1, 2) H(3) CX(3, 4) CX(3, 2) CX(1, 0) H(1) CX(5, 4) H(5)
M(0) M(1) M(4) M(5)
(c[4] == 1 ? X(2) X(3))
(c[5] == 1 ? Z(3))
(c[0] == 1 ? X(2))
(c[1] == 1 ? Z(2) Z(3))
```

Let $I_{QGT}$ be the set of initial states for Quantum Gate Teleportation. It consists of only one initial state as follows:

```
{(isEnd: false)
(prob: 1)
(qstate: (q[0]: a . |0> + b . |1>)
         (q[1]: |0>) (q[2]: |0>)
         (q[3]: |0>) (q[4]: |0>)
         (q[5]: c . |0> + d . |1>))
(bits: empty)
(actions:
```

```
H(1) CX(1, 2) H(3) CX(3, 4) CX(3, 2) CX(1, 0) H(1) CX(5, 4) H(5)
M(0) M(1) M(4) M(5)
(c[4] == 1 ? X(2) X(3))
(c[5] == 1 ? Z(3))
(c[0] == 1 ? X(2))
(c[1] == 1 ? Z(2) Z(3))}
```

where `a`, `b`, `c`, and `d` are Maude constants denoting arbitrary scalars such that $|a|^2 + |b|^2 = |c|^2 + |d|^2 = 1$. Initially, the `isEnd` observable component is false, the `prob` observable component is one, the `qstate` is a symbolic state that is the same as the input state of the protocol, the `actions` observable component contains the action list describing how the protocol works.

### Model checking quantum gate teleportation

Let $K_{QGT}$ and `init` be the Kripke structure and the initial state for Quantum Gate Teleportation, respectively. To model check that $K_{QGT}$ satisfies desired properties, we specify $A_{QGT}$ and $L_{QGT}$. $A_{QGT}$ has one atomic proposition `isSuccess`. $L_{QGT}$ is specified as follows:

```
eq {(isEnd: true) (qstate: Q) (prob: Prob) OCs} |= isSuccess
= Prob > 0 implies (qubitAt(Q, 3 2) == qubitAt(targetQState, 0 1) or
                   (-1) . qubitAt(Q, 3 2) == qubitAt(targetQState, 0 1)) .
eq {OCs} |= PROP = false [owise] .
```

where `Q` and `Prob` are Maude variables denoting the whole quantum state and the probability at a state, respectively. `targetQState` represents the outcome of the protocol, the controlled-NOT gate of the two arbitrary states, which is defined as follows

```
eq targetQState = ((q[0]: c . |0> + d . |1>) (q[1]: a . |0> + b . |1>)).CX(0, 1)
```

The two equations to specify $A_{QGT}$ say that `isSuccess` holds at a state if the state contains `(isEnd: true)`, `(qstate: Q)`, and `(prob: Prob)` such that the condition `qubitAt(Q, 3 2) == qubitAt(targetQState, 0 1)` or `(-1) . qubitAt(Q, 3 2) == qubitAt(targetQState, 0 1)` holds whenever `Prob > 0`, meaning that Alice can successfully teleport two arbitrary unknown quantum states through the controlled-NOT gate to Bob at the end with a non-zero probability. Again, because of the *global phase* in quantum states as shown in *Nielsen & Chuang (2010)*, we use `(-1) . qubitAt(Q, 3 2) == qubitAt(targetQState, 0 1)` in addition to `qubitAt(Q, 3 2) == qubitAt(targetQState, 0 1)` in the condition. Let `gateProp` be an LTL formula defined as `<> isSuccess`.

We want to model check that $K_{QGT} = \langle S, I_{QGT}, T, A_{QGT}, L_{QGT} \rangle$ satisfies `gateProp` from the initial state `init` in Maude as follows:

```
red modelCheck(init, gateProp) .
```

No counterexample is found in just 176 ms and so $K_{QGT}$ satisfies `gateProp`. In other words, we successfully verify the correctness of Quantum Gate Teleportation by using our symbolic model checking approach.

## REMARK ON QUANTUM GATE TELEPORTATION

This section describes a remark on Quantum Gate Teleportation where we show that a circuit for the original version of the protocol does not satisfy its desired property, while

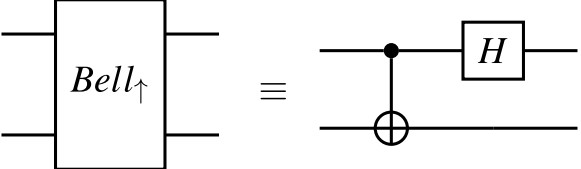

**Figure 6** A Bell-basis measurement gate.

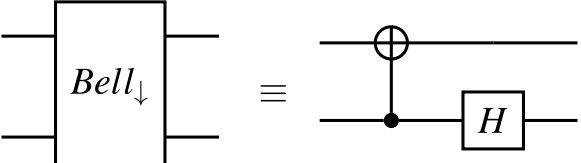

**Figure 7** A Bell-basis measurement gate inserted upside down.

we present a revised circuit for the protocol and verify that it satisfies its desired property using our symbolic model checking approach.

The reader can refer to the circuit for the original version of Quantum Gate Teleportation from *Gottesman & Chuang (1999)*, Fig. 2. In that figure, they use a Bell measurement denoted by the box $B$ twice in the circuit. A Bell measurement can be described in terms of a **CX** gate, a Hadamard gate, and a measurement in the standard basis. However, there are two possible orientations for applying the **CX** gate and the Hadamard gate. The difference is apparent by comparing Figs. 6 and 7 as also presented by *Williams (2008)*. The Bell measurement that uses the Bell-basis measurement gate in Fig. 6 is called the up Bell measurement. Meanwhile, the Bell measurement that uses the Bell-basis measurement gate inserted upside down in Fig. 7 is called the down Bell measurement. *Gottesman & Chuang (1999)* did not explicitly clarify which Bell measurement was used. However, they described it in their article exactly as follows:

> The box $B$ represents measurement in the Bell basis; that is, if the two qubits entering $B$ are found to be $|00\rangle + |11\rangle$ (leaving out the $\sqrt{2}$ normalization for clarity), then the outputs $xy = 00$; for $|01\rangle + |10\rangle$, $xy = 10$; for $|00\rangle - |11\rangle$, $xy = 01$; and for $|01\rangle - |10\rangle$, $xy = 11$.

Note that $xy$ denotes the first and second qubits entering the box $B$. Based on the above description in the original article, we use the down Bell measurement. If so, the circuit for Quantum Gate Teleportation is depicted in Fig. 8. Let us call this circuit the original circuit.

We conducted a model checking experiment with the same initial state and the desired property described in 'Quantum Gate Teleportation' for the original circuit. A counterexample was found by the Maude LTL model checker, implying that Quantum Gate Teleportation does not satisfy the property with the use of the original circuit. Because

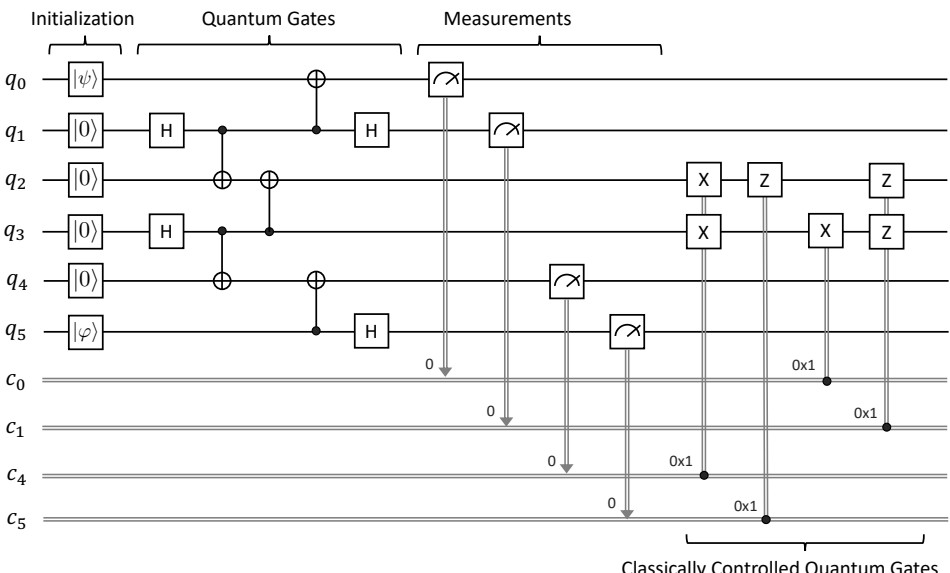

**Figure 8** A possible circuit for Quantum Gate Teleportation based on the description from *Gottesman & Chuang (1999)*.

the counterexample is overlong, we just excerpt and present the final state repeated forever in the counterexample as follows:

```
{(isEnd: true)
(prob: 1/16)
(qstate:
    (q[0]: |0>) (q[1]: |0>)
    (q[4]: |0>) (q[5]: |1>)
    (q[3 2]: (a .* c) . |0> (x) |0> + (a .* d) . |1> (x) |1> +
             (b .* c .* -1) . |0> (x) |1> + (b .* d .* -1) . |1> (x) |0>)
(bits: (0 |-> 0, 1 |-> 0, 4 |-> 0, 5 |-> 1))
(actions: nil)
```

We can see that the qubits at indices 3 and 2 do not match the controlled-NOT gate of the two arbitrary states represented by targetQState in 'Quantum Gate Teleportation (QGT)'. That is why the original circuit for Quantum Gate Teleportation does not satisfy the property. To replay the counterexample, we can conduct as follows: the sequence of quantum gates in the original circuit is applied as usual for the initial state, followed by the measurements of the first, second, fourth, and fifth qubits such that their measurement outcomes are 0, 0, 0, and 1, respectively. Furthermore, even if we use the up Bell measurement in place of the down Bell measurement, its corresponding circuit of Quantum Gate Teleportation also does not satisfy the property. Therefore, regardless of the use of either up or down Bell measurement, the quantum circuit proposed by *Gottesman & Chuang (1999)* does not enjoy the desired property.

We revised the original circuit for Quantum Gate Teleportation by changing the positions of the $Z$ and $X$ gates applied to qubits with respect to the values of $c_5$ and $c_0$, respectively. Concretely, we apply the $Z$ gate to $q_3$ instead of $q_2$, the $X$ gate to $q_2$ instead of $q_3$ depending on the values of $c_5$ and $c_0$, respectively, for the revised circuit compared to the original

**Table 3  Experimental results.**

| Protocol | Qubits | States | Verification time |
|---|---|---|---|
| Superdense coding | 2 | 16 | 1 ms |
| Quantum teleportation | 3 | 27 | 3 ms |
| Quantum secret sharing | 4 | 65 | 13 ms |
| Entanglement swapping | 4 | 29 | 3 ms |
| Two mirror-image teleportation | 6 | 151 | 15 ms |
| Quantum gate teleportation | 6 | 168 | 176 ms |
| Quantum network coding | 10 | 7,373 | 2,446 ms |

circuit. The revised circuit is depicted in Fig. 5 in 'Quantum Gate Teleportation (QGT)'. For the revised circuit of Quantum Gate Teleportation, we successfully verified the correctness of the protocol by using our symbolic model checking approach. This demonstrates the usefulness of our symbolic model checking approach for verifying quantum circuits.

## EXPERIMENTAL RESULTS

This section summarizes our experimental results for verifying the correctness of several quantum communication protocols with our symbolic model checking approach, including Superdense Coding introduced by *Bennett & Wiesner (1992)*, Quantum Teleportation introduced by *Bennett et al. (1993)*, Quantum Secret Sharing introduced by *Hillery, Bužek & Berthiaume (1999)*, Entanglement Swapping introduced by *Zukowski et al. (1993)*, Quantum Gate Teleportation introduced by *Gottesman & Chuang (1999)*, Two Mirror-image Teleportation introduced by *Williams (2008)*, and Quantum Network Coding introduced by *Satoh, Gall & Imai (2012)*. The experiments were conducted with an iMac that carries a 4 GHz microprocessor with eight cores and 32 GB memory of RAM. The experimental results are shown in Table 3. The second, third, and fourth columns denote the number of qubits in each protocol, the number of states in the reachable state space of each protocol under model checking, and the verification time for each protocol, respectively.

For case studies ranging from two to ten qubits, model checking experiments were quickly completed in times from 1 ms to 2,446 ms as shown in Table 3. The number of states in the reachable state space for Quantum Network Coding with ten qubits is notably larger compared to the number of states in the reachable state space for each of the first six protocols. Nevertheless, the model checking experiment for Quantum Networking Coding could be completed in a short amount of time. Without the aid of computer programs, such as our support tool implemented in Maude, it is almost impossible to achieve the same results. These results demonstrate the usefulness of our symbolic model checking approach to verifying quantum circuits in Maude. As one piece of future work, we would like to tackle more case studies with a larger number of qubits to present the scalability of our approach.

# DISCUSSION

This section discusses some limitations of our approach, challenges in using a classical model checker to verify quantum circuits with their desired properties, and how we address these challenges.

## Limitations

In the context of symbolic reasoning for complex numbers, we have extended rational numbers supported in Maude to deal with complex numbers. Our objective is to represent arbitrary complex numbers in a pure quantum state using fresh constants representing arbitrary complex numbers and manipulating them without using any concrete values for real numbers. As a result, our current framework cannot handle any concrete values for real numbers. Nevertheless, we plan to explore the use of float numbers supported in Maude for simulating quantum circuits with concrete values in the future. As shown in Appendix 2, we have specified some basic operations for complex numbers, such as multiplication, division, addition, conjugation, absolute, power, and square roots. Note that the formal specification of complex numbers is not complete in this article. Hence, there may be some cases where symbolic reasoning for complex numbers could not further reduce terms. As part of our future work, we aim to enrich the framework for complex number reasoning as much as possible.

Regarding symbolic reasoning for quantum computation, we only support a limited set of quantum gates, including *I*, *X*, *Y*, *Z*, *H*, *CX*, *S*, *T*, *CY*, *CZ*, *SWAP*, *CCY*, *CCZ*, and *CSWAP* gates. Consequently, a restricted set of quantum protocols can be described in our framework. Although we support a universal set of quantum gates, including the Clifford gates (*i.e.,* *H*, *S*, and *CX*) and the phase shift gate *T*, universal quantum computations could not be handled by our framework at this moment because the symbolic reasoning for complex numbers is not complete. We would like to extend our symbolic reasoning to handle more quantum gates so that a wider range of quantum protocols can be verified using our approach.

## Challenges and future prospects

There are some challenges that we need to address in order to use the Maude LTL model checker, a classical model checker, to verify quantum circuits with their desired properties. In addition, this section also outlines future prospects in model checking quantum circuits.

- First, we need to devise a way to specify quantum states, quantum gates, and measurements in a Maude specification to reason about quantum computation. We specified quantum states, quantum gates, and measurements in Dirac notation and used a set of laws from quantum mechanics and basic matrix operations to reason about quantum computation automatically in Maude. Moreover, Maude does not support complex numbers as a built-in type. Therefore, we extended rational numbers, a built-in type in Maude, so as to deal with complex numbers symbolically as described in Appendix A.
- Second, quantum gates can be applied to quantum states in a deterministic way, while quantum measurements are inherently non-deterministic and the states after

the measurements will collapse. It is natural to use rewriting logic to describe non-deterministic or concurrent behaviors in Maude in the form of rewrite rules. We specified a binary projective measurement using two rewrite rules corresponding to two non-deterministic choices, as described in 'Formal Specification'.

- Third, we need to appropriately handle both pure and entangled states in our formal specification and devise a simple notation to conveniently describe the behavior of quantum circuits. We specified a whole quantum state as a collection of qubits associated with indices in quantum circuits, enabling flexible reference to specific parts of a quantum state using indices. The behavior of quantum circuits has been specified as a list of actions, a convenient and sufficient approach for us to concisely describe their behavior.

- Lastly, we need to represent quantum states and quantum gates in order to effectively model check quantum circuits with as many qubits as possible. Using Dirac notation in our specification allows us to avoid many redundancies compared to explicitly using vectors and matrices to represent quantum states and quantum gates, respectively, making our representation more compact than that of *Paykin, Rand & Zdancewic (2017)*. Early work proposed by *Gay, Nagarajan & Papanikolaou (2005)* could not support the analysis of quantum systems with five qubits, while our approach could handle case studies of up to ten qubits, showing the effectiveness of our approach. However, in order to handle quantum systems with hundreds of qubits in the future, we need to use or come up with advanced techniques to effectively simulate quantum computation(*e.g.*, using decision diagrams for quantum computing proposed by *Wille, Hillmich & Burgholzer (2022)*, *Wille, Hillmich & Burgholzer (2023)*) or analyze such quantum systems in a modular way.

Last but not least, our approach implemented in Maude can be a first step toward a general framework for specifying and verifying quantum circuits when we can reuse some essential elements in the Kripke structures. For specifying and verifying a quantum circuit, we are supposed to define an initial state, describe the behavior of the quantum circuit in terms of a list of actions, and specify atomic propositions and the labeling function based on which a desired property can be constructed. Given the system specification with the initial state and the desired property, the Maude LTL model checker automatically checks whether the system specification satisfies the desired property reachable from the initial state.

## RELATED WORK

There are several studies in the early work of formal specification and verification of quantum protocols, such as *Gay, Nagarajan & Papanikolaou (2005)*; *Elboukhari, Azizi & Azizi (2010)*. For example, *Gay, Nagarajan & Papanikolaou (2005)* provide a way to use classical model checkers (*e.g.*, PRISM - a probabilistic model checker) to analyze quantum protocols. They give each quantum state a unique number and the transition from a unique number to another unique number models the action of quantum gates and measurements. Their approach needs to enumerate states, calculate the state transitions in advance, and then encode them into a PRISM specification. Although they developed

a so-called PRISMGEN tool to automate this, their approach is impractical in reality and only supports two or three qubits because of the exponential growth of the number of states. Our approach does not need to enumerate such states in advance because a quantum state is directly specified in Dirac notation with scalars. Moreover, rewrite rules are used to specify the action of quantum gates and measurements, making our approach feasible to deal with more qubits. For example, we have verified the correctness of Quantum Network Coding that has ten qubits by using our symbolic model checking approach.

*Ying (2021)* proposes a framework for assertion-based verification of quantum circuits by using model checking techniques. In this work, quantum circuits are represented by tensor networks, where a tensor is a multi-dimensional array of complex numbers, and two tensors sharing indices are connected by a tensor contraction, which basically is matrix calculation. Quantum states and quantum gates are specified as tensors, and quantum circuits are specified as tensor networks. Given a quantum state as an input to a quantum circuit, the output will be the contraction of the quantum state and the quantum circuit. Assertions or properties about quantum circuits are specified using computation tree quantum logic (CTQL), an extension of the Birkoff-von Neumann quantum logic presented in *Birkhoff & Neumann (1936)*. Using tensor network representation of quantum circuits, they can conveniently implement a reachability analysis algorithm and a model checking algorithm for quantum circuits by contraction of tensor networks. Compared to our work, Dirac notation is used to express quantum states and quantum gates instead of tensors, and quantum circuits are described by an action list with simple notations instead of tensor networks. Our reasoning on quantum circuits is mainly based on the laws of quantum mechanics and matrix operations with Dirac notation, but they construct contraction between tensors in tensor networks. We use LTL to express desired properties for quantum circuits instead of CTQL. It seems that they cannot deal with quantum circuits together with the appearance of classical bits obtained from measurements, while our approach can do so. Moreover, they do not show any case study to which their framework can apply.

*Burgholzer & Wille (2021)* have proposed an advanced method for equivalence checking of quantum circuits. Their approach involves two quantum circuits $G$ and $G'$ as inputs and they check whether the two quantum circuits are equivalent. They leverage two key observations: (1) quantum circuits are inherently reversible, and (2) even small differences in quantum circuits may impact the overall behavior of quantum circuits. Let us suppose that two quantum circuits are sequences of unitary transformations: $G = U_{m-1} \ldots U_0 = U$ and $G' = U'_{m'-1} \ldots U'_0 = U'$ operating on $n$ qubits. Executing a quantum circuit to evolve an initial state $|\psi\rangle$ to another state $|\psi'\rangle$ such that $U_m \ldots U_0 |\psi\rangle = U|\psi\rangle = |\psi'\rangle$ is called *simulation*. For (1), $G$ is equivalent to $G'$ if and only if $(U'_0)^{-1} \ldots (U'_{m'-1})^{-1} U_{m-1} \ldots U_0 = I$ or $(U'_0)^\dagger \ldots (U'_{m'-1})^\dagger U_{m-1} \ldots U_0 = I$ when $(U'_i)^{-1} = (U'_i)^\dagger$ due to their unitary matrices. They employ decision diagrams to represent matrices and try to resolve $(U'_{i'})^\dagger U_i$ into the identity matrix $I$ for effectively solving the equivalence checking problem. For (2), comparing the entire matrices of $U$ and $U'$ is unnecessary when two quantum circuits are not equivalent. Comparing some columns of each $U$ and $U'$ is enough to conclude the equivalence checking problem. Constructing a single column of $U$ (or $U'$) equates to simulating $G$ (or $G'$) with the standard basis state $|i\rangle$ as follows: $|u_i^0\rangle = U_0|i\rangle$, $|u_i^{(j)}\rangle = U_j \cdot u_i^{(j-1)}$ for $j \in \{1, \ldots, m-1\}$.

As the results of these simulations, if $|u_i\rangle = |u_i^{(m-1)}\rangle$ and $|u_i'\rangle = |u_i'^{(m'-1)}\rangle$ are different, it indicates non-equivalence of the two quantum circuits. These can be quickly checked through a randomized selection of some columns with simulations. While their approach is promising, it differs from ours when we take a formal specification for a quantum circuit and a formal property for a desired property as inputs and check whether the quantum circuit satisfies the desired property. Nevertheless, we may utilize their idea to extend our symbolic reasoning to check the equivalence of quantum circuits, which would be one interesting direction.

The ZX calculus, as proposed by *Coecke & Duncan (2011)*, is a graphical formal language for quantum systems equipped with a robust set of rewrite rules that enable a graphical rewriting system for quantum computation. The graphical formalism of the ZX calculus can be implemented in the automated rewriting system Quantomatic proposed by *Kissinger & Zamdzhiev (2015)* for the automatic simplification process. The ZX calculus has various applications in quantum computing, such as verifying quantum error-correcting codes and equivalence checking of quantum circuits. For example, *Peham, Burgholzer & Wille (2022)* proposed an approach to the equivalence checking of quantum circuits using the ZX calculus. Given two quantum circuits $U$ and $U'$, they produce their corresponding representations as ZX-diagrams $D$ and $D'$. These diagrams are then combined into $D^\dagger D'$ and simplified using the set of rewrite rules. If the result is in the form of the identity diagram, they can conclude their equivalence. Otherwise, nothing can be concluded because there are multiple forms for a ZX diagram in general. This approach is intuitive when we can see which rewrite rules are used and how ZX diagrams are changed accordingly. Our approach based on Dirac notation may be less intuitive. However, our approach is to verify whether quantum circuits satisfy their desired properties, which are different from the equivalence checking of quantum circuits as mentioned above.

*Rand, Paykin & Zdancewic (2018)* implement the QWIRE programming language, a high-level abstraction to describe quantum circuits for programmers, in the Coq proof assistant and use Coq's theorem proving features to prove desired properties for quantum circuits. They explicitly use matrix representations, while we use Dirac notation to reason about quantum circuits. As the inherent problem of theorem proving, they need to provide necessary lemmas in order to prove some properties that can be considered the most challenging task in theorem proving. Our approach is model checking and so it is completely automatic.

Our symbolic approach to model checking quantum circuits is inspired by *Shi et al. (2021)* and so it is the closest work to ours. However, their approach is oriented to theorem proving, not model checking. They also use Dirac notation with a small set of laws to specify quantum states, quantum gates, measurements, and reasoning about quantum circuits in Coq, an interactive theorem prover. However, they usually require human users to provide necessary lemmas to complete their proofs, which is generally a challenging task. Meanwhile, our approach is fully automatic and requires no human intervention. Moreover, our implementation can be a first step toward a general framework to formally specify and verify quantum circuits in a symbolic way in Maude.

## CONCLUSION

We have proposed a symbolic approach to model checking quantum circuits using a set of laws from quantum mechanics and basic matrix operations with Dirac notation. We have analyzed the correctness of several quantum communication protocols in the early stage of quantum communication: Superdense Coding, Quantum Teleportation, Quantum Secret Sharing, Entanglement Swapping, Quantum Gate Teleportation, Two Mirror-image Teleportation, and Quantum Network Coding as case studies to demonstrate the usefulness of our approach. In particular, we have identified that the original version of Quantum Gate Teleportation did not satisfy its desired property, and have proposed a revised version and confirmed its correctness using our approach and support tool. Moreover, our implementation developed in Maude can be a first step toward a general framework to formally specify and verify quantum circuits using our symbolic model checking approach. Our specification considers the probabilities of measurements, and then we can tackle both qualitative and quantitative properties with the built-in LTL model checker in Maude.

As one piece of our future work, we would like to extend our symbolic reasoning to handle more quantum gates and more complicated reasoning on complex number operations. As usual, we need to conduct more case studies to demonstrate the usefulness of our approach/implementation. As another line of future work, we also would like to apply our symbolic approach to model checking quantum programs and quantum cryptography protocols.

## ACKNOWLEDGEMENTS

The authors wish to thank the anonymous reviewers who commented on drafts of this article.

## APPENDIX

### A. Symbolic reasoning for scalars

This section describes how we extend the built-in rational numbers in Maude to handle scalars. Please note that scalars and complex numbers are used interchangeably in this article. Some operators for scalars are specified to tackle case studies in this article, including multiplication, division, addition, conjugation, absolute, power, and square roots. In the sequel, the Maude syntax is used in our description.

In addition to the built-in sort `Rat` of rational numbers, we introduce two sorts `Real` and `Complex` representing real numbers and complex numbers, respectively, as follows:

```
sorts Real Complex .
subsort Rat < Real < Complex
```

where `Rat` is a sub-sort of `Real` and `Real` is a sub-sort of `Complex`. This indicates that rational numbers are a subset of real numbers, and real numbers are a subset of complex numbers. It is worth noting that the built-in sort `Nat` of natural numbers is a sub-sort of `Rat`. While it would be useful to use real numbers to simulate quantum circuits with concrete values (*e.g.,*

$\pi$), it is not the goal of this study. Our objective is to represent arbitrary complex numbers in a pure quantum state using fresh constants of sort `Complex` and manipulating them without using any concrete values for real numbers. As a result, our focus is on specifying complex numbers with rational numbers for conducting model checking experiments for case studies in this article. The specification of concrete real numbers is planned as part of our future work.

We first define an operator representing the imaginary unit for complex numbers as follows:

```
op i : -> Complex [ctor] .
```

where `i` serves as a constructor of the imaginary unit for complex numbers with the `ctor` attribute.

We then define some operators for complex numbers, including multiplication, division, addition, and conjugation, as follows:

```
op _.*_  : Complex Complex -> Complex [comm assoc prec 32] .
op _./_  : Complex Complex -> Complex [prec 31] .
op _.+_  : Complex Complex -> Complex [comm assoc prec 33] .
op (_)^* : Complex         -> Complex [prec 30] .
```

where `_` represents arguments of sort `Complex` for the operators defined. The `_.*_` and `_.+_` operators satisfy commutativity and associativity with the `comm` and `assoc` attributes. Each operator is assigned a different precedence, indicated by the `prec_` attribute, with `_` denoting a parameter representing a numerical precedence value.

We next define some operators for complex numbers to represent absolute, power, and square roots as follows:

```
op Abs  : Complex     -> Real .
op Pow  : Complex Rat -> Complex .
op Sqrt : Complex.    -> Complex .
```

Now we are ready to define the semantics of the operators introduced above through equations. Because of the self-explanation of equations, we do not explain each equation in detail for the sake of brevity. We defined some Maude variables before using them in equations as follows: `N` is a Maude variable of sort `Nat`; and `PR'` are Maude variables of sort `PosRat` (for positive rational numbers); `R`, `R1`, and `R2` are Maude variables of sort `Rat`; and `C`, `C1`, `C2`, and `C3` are Maude variables of sort `Complex`.

We first define the semantics of multiplication, division, and addition for complex numbers, particularly focusing on cases involving rational numbers.

```
eq R1 .* R2 = R1 * R2 .
eq R1 ./ R2 = R1 / R2 .
eq R1 .+ R2 = R1 + R2 .
```

where `*`, `/`, and `+` are built-in operators for multiplication, division, and addition of rational numbers if applicable.

The properties of the imaginary unit are defined as follows:

```
eq i .* i = -1 .
eq Abs(i) = 1 .
```

The semantics of multiplication is defined as follows:

```
eq C .* 1 = C .
eq C .* 0 = 0 .
eq C .* (C)^* = Pow(Abs(C), 2) .
ceq C .* (1 ./ C) = 1 if not C :: Rat .
```

where `C :: Rat` to check whether `C` belongs to the sort of `Rat`.

The semantics of the division is defined as follows:

```
ceq 1 ./ (1 ./ C) = C if C =/= 0 .
ceq 1 ./ (C1 .* C2) = (1 ./ C1) .* (1 ./ C2)
if not C1 :: Rat or-else not C2 :: Rat .
```

The semantics of the addition is defined as follows:

```
eq C .+ 0 = C .
```

We would like to construct normal forms for complex numbers with the existence of multiplication, division, and addition operators.

Therefore, some equations are defined as follows:

```
--- multiplication distributes over addition
eq C1 .* (C2 .+ C3) = C1 .* C2 .+ C1 .* C3 .
--- constructing normal forms for addition
ceq C .* R1 .+ C .* R2 = C .* (R1 .+ R2) if not C :: Rat .
ceq C .+ R .* C = C .* (R .+ 1) if not C :: Rat .
ceq C .+ C = C .* 2 if not C :: Rat .
--- constructing normal forms for division
ceq C1 ./ C2 = C1 .* (1 ./ C2) if C1 =/= 1 .
```

The absolute of a positive number is a positive number and so we define it as follows:

```
eq Abs((Abs(C))) = Abs(C) .
eq Abs((Pow(Abs(C), N))) = Pow(Abs(C), N) .
eq Abs(R) = abs(R) .
```

where `abs` is a built-in operator for the absolute of rational numbers if applicable.

The semantics of square roots is defined as follows:

```
eq Sqrt(1) = 1 .
eq Sqrt(0) = 0 .
eq Sqrt(PR) .* Sqrt(PR) = PR .
eq 1 ./ Sqrt(PR) .* 1 ./ Sqrt(PR) = 1 ./ PR .
eq PR .* (1 ./ Sqrt(PR)) = Sqrt(PR) .
eq Sqrt(PR / PR') = Sqrt(PR) .* (1 ./ Sqrt(PR')) .
eq Sqrt(PR) .* (R / PR) = R .* (1 ./ Sqrt(PR)) .
```

It is worth noting that the semantics of square roots is partially implemented for positive rational numbers in this article, while we leave others as part of our future work. This decision stems from the sufficiency of using square roots of some positive rational numbers for the case studies used in this article.

The semantics of power is defined as follows:

```
eq Pow(R1, R2) = (R1)^(R2) .
```

where `(.)(.)` is a built-in operator for the power of rational numbers if applicable.

Lastly, the semantics of conjugate is defined as follows:

```
eq (Sqrt(PR))^* = Sqrt(PR) .
eq (C1 .* C2)^* = (C1)^* .* (C2)^* .
eq (C1 ./ C2)^* = (C1)^* ./ (C2)^* .
eq (C1 .+ C2)^* = (C1)^* .+ (C2)^* .
eq (R)^* = R .
```

Based on what we defined above, we can symbolically reason on complex numbers with rational numbers for our case studies in this article.

## B. Quantum computation with our specifications

Although many textbooks on quantum mechanics cover the application of quantum operations to quantum states, this section is specifically dedicated to explaining how these operations apply to quantum states with respect to the specifications presented in this article.

It is possible to treat the whole quantum state uniformly. However, we do not take this way; instead, we specify quantum states as a collection of qubits associated with indices that start from 0 to $N - 1$, where $N$ is the total number of qubits, as described in 'A generic maude specification of quantum circuits'. Our way has some advantages as follows:

- We can flexibly refer to a specific part of a quantum state using indices. This is very helpful when we want to take a part of the whole quantum state to check whether it satisfies certain conditions. For example, in Quantum Teleportation protocol, we need to verify whether the third qubit at the end is equal to the first qubit at the beginning.
- When we apply a quantum gate to the whole quantum state at certain indices, if the indices of the whole quantum state belong to an isolated part, we can perform a local computation by considering only that part and leaving other parts unchanged. This may make the computation faster, especially for a large number of qubits.

### How quantum gates are applied to the whole quantum state

Let us suppose that we want to apply a single-qubit quantum gate $X$ (*i.e.*, $B1 + B2$) to the whole quantum state at index $k$, and the whole quantum state contains $(q[i, \ldots, k, \ldots, j] : |\psi\rangle)$. First, we need to prepare the cylindrical extension $U$ of $X$ to make it have the same dimension as $|\psi\rangle$ as follows:

$$U = I \otimes \cdots \otimes X \otimes \cdots \otimes I = I \otimes \cdots \otimes B1 \otimes \cdots \otimes I + I \otimes \cdots \otimes B2 \otimes \cdots \otimes I.$$

Then, the whole quantum state after applying the quantum gate $X$ will become to contain $(q[i, \ldots, k, \ldots, j] : U \times |\psi\rangle)$. Note that other quantum states associated with indices except for $i, \ldots, k, \ldots, j$ are not affected and remain unchanged. Therefore, the way we use to represent the whole quantum state allows us to perform local computations where only some parts are considered, while other parts remain unchanged.

Let us suppose that we want to apply a two-qubit quantum gate $CX$ (*i.e.*, $B_0 \otimes I + B_3 \otimes X$) to the whole quantum state at indices $k$ and $l$, and the whole quantum state contains $(q[i, \ldots, k, \ldots, j] : |\psi_1\rangle)$ $(q[i', \ldots, l, \ldots, j'] : |\psi_2\rangle)$. First, we need to combine two quantum states $|\psi_1\rangle$ and $|\psi_2\rangle$ so that the whole quantum state will become to contain $(q[i, \ldots, k, \ldots, j, i', \ldots, l, \ldots, j'] : |\psi_1\rangle \otimes |\psi_2\rangle)$. Second, we need to prepare the cylindrical extension $U$ of $CX$ to make it have the same dimension as $|\psi_1\rangle \otimes |\psi_2\rangle$ as follows:

$$U = I \otimes \cdots \otimes B_0 \otimes \cdots \otimes I \otimes \cdots \otimes I + I \otimes \cdots \otimes B_3 \otimes \cdots \otimes X \otimes \cdots \otimes I.$$

Finally, the whole quantum state after applying the quantum gate $CX$ will become to contain

$(q[i, \ldots, k, \ldots, j, i', \ldots, l, \ldots, j'] : U \times (|\psi_1\rangle \otimes |\psi_2\rangle))$.

Note that other quantum states associated with indices except for $i, \ldots, k, \ldots, j, i', \ldots, l, \ldots, j'$ are not affected and remain unchanged.

The procedure for applying a three-qubit quantum gate to the whole quantum state is similar to that for a two-qubit quantum gate. Therefore, it can also be done.

## How to detect single and entangled states

We develop some heuristics for detecting single and entangled qubit states from which they are separated from each other.

Let us suppose that the whole quantum state contains $q[i, \ldots, k, \ldots, j] : |\psi\rangle$, and we want to perform a measurement on the whole quantum state at index $k$. After the measurement, the whole quantum state will become to contain $(q[i, \ldots, j] : |\psi'\rangle) \, (q[k] : |\phi\rangle)$, where the quantum state at index $k$ is separated into $|\phi\rangle$ whose value either $|0\rangle$ or $|1\rangle$ and the quantum state at the other indices becomes $|\psi'\rangle$ according to the result of the measurement.

Furthermore, if the whole quantum state contains $(q[i, \ldots, k, \ldots, j] : |\psi_1\rangle \otimes |\phi\rangle \otimes |\psi_2\rangle)$ where $|\phi\rangle$ is either $|0\rangle$ or $|1\rangle$, we can also detect it. In this case, the whole quantum state will become to contain $(q[i, \ldots, j] : |\psi_1\rangle \otimes |\psi_2\rangle) \, (q[k] : |\phi\rangle)$. Additionally, we can separate the tensor product of two Bell states as well. Although these heuristics do not guarantee coverage of all cases, they are sufficient for the case studies used in this article. We aim to improve our heuristics to cover as many cases as possible in our future work.

### Funding

This work was supported by JST SICORP Grant Number JPMJSC20C2, Japan and JSPS KAKENHI Grant Numbers JP23H03370, JP23K19959, JP24K20757. The funders had no role in study design, data collection and analysis, decision to publish, or preparation of the manuscript.

### Grant Disclosures

The following grant information was disclosed by the authors:
JST SICORP: JPMJSC20C2.
JSPS KAKENHI: JP23H03370, JP23K19959, JP24K20757.

### Competing Interests

The authors declare there are no competing interests.

### Author Contributions

- Canh Minh Do conceived and designed the experiments, performed the experiments, analyzed the data, performed the computation work, prepared figures and/or tables, authored or reviewed drafts of the article, and approved the final draft.
- Kazuhiro Ogata conceived and designed the experiments, analyzed the data, authored or reviewed drafts of the article, and approved the final draft.

### Data Availability

The raw data (implementation, case studies, and code) are available at Zenodo: Canh Minh Do. (2024). canhminhdo/QTC-Maude: QTC-Maude v1.0.2 (v1.0.2). Zenodo. https://doi.org/10.5281/zenodo.10783951.

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
