# Peer review of "Symbolic model checking quantum circuits in Maude"

_PeerJ Computer Science, doi:10.7717/peerj-cs.2098_

## Round 0.1 · original submission · Major Revisions

Dear authors,

Reviewers have looked at your manuscript. As you can see, they find your work interesting, but also have quite a few suggestions for improvement, in particular describing background and context for the work, which I think are all valuable remarks, therefore I follow their recommendations and ask you to make a major revision of your paper. Please address the comments by the reviewers.

Reviewer 1 ·

Basic reporting

In Quantum Computing, there are two different ways to represent and manipulate quantum states: using linear algebra quantum computing model or continuous quantum computing model. The paper under review aims to model check quantum circuits specified in linear algebra quantum computing, where the quantum states are represented as complex-valued vectors in a Hilbert space and quantum gates are represented as unitary matrices that act on these vectors. The state space for these circuits is finite and therefore it can be explored and checked using a classic model checker.

In this paper, the authors describe a Maude specification for the linear algebra quantum computing model and use the Maude internal LTL model checker to check some properties of several well-known quantum circuits.

The paper is relatively well-written. The Maude specification of the quantum computing model is well-described and the quantum circuit examples are explained in detail.
However, the reader has to be familiar with Maude to completely understand the specification.

Experimental design

The declared goal of the paper is to present "a symbolic approach to model checking quantum circuits using a set of laws from quantum mechanics", but is not very clear what kind of properties will be investigated, why these properties are interesting from a quantum point of view, which are the main challenges to check these properties, and why only quantum circuits and not quantum programs in general.

Validity of the findings

The current version is a very nice exercise of using Maude for naively analyzing quantum circuits. Analyzing quantum circuits is not a complicated task, many times it can be done manually. Moreover, there exist some previous approaches using probabilistic model-checkers (e.g., Prism).
What is different from using the current approach?
Therefore I suggest to the author to consider quantum programs written in a quantum programming language, e.g., Q#.

The Maude specification is publicly available, I tested several examples and they worked as described in the paper. For some examples, it is not necessary to use LTL model checker, the use of the bounded model checker (the search command) is enough.

Additional comments

Detailed comments

p. 2, r. 79-80: "We demonstrate that our approach/implementation can be used as a general framework to formally
specify and verify quantum circuits."
To have such a framework, much more has to be done: define a specification language suitable for quantum programs/circuits, define a suitable operational semantics for quantum programs/circuits, formally define when a given quantum program/circuit satisfies a given specification, and how all these are soundly captured by the proposed framework. There are some challenges in using classical model checkers to analyze quantum programs/circuits and the paper should explicitly show how these challenges are addressed.

p. 2, r. 87: "the quantitative and qualitative properties" should be formally defined later in the paper.

p. 3., r. 99-100: "Quantum Gate Teleportation does not satisfy its
desired property under our conjecture using our approach and support tool." - What is the "standard" specification for the QGT? What is the relationship between this specification and that used in the paper?

2.1 Basic Quantum Mechanics - Mention that the linear algebra approach is used.

3 SYMBOLIC REASONING

"The “symbolic” word means that we use bra-ket notation instead of explicitly complex vectors and
200 matrices as Paykin et proposed in Paykin et al. (2017), which makes our representations more compact."

The algebraic specification of complex numbers is not new and cannot be mentioned as a contribution. See, e.g.,

Bergstra, Jan & Tucker, John. (2006). Elementary Algebraic Specifications of the Rational Complex Numbers. 459-475. 10.1007/11780274_24.
and
Joseph A. Goguen, José Meseguer:
Order-Sorted Algebra I: Equational Deduction for Multiple Inheritance, Overloading, Exceptions and Partial Operations. Theor. Comput. Sci. 105(2): 217-273 (1992)

I am sure that there also other Maude specifications for them as well.
The use of the word "symbolic" in the title must be revised, the model-checking approach from this version is not symbolic.

p. 5, r. 206-207: "Some constructors for scalars, such as multiplication, fraction ... are formalized." - What exactly does mean that? "multiplication, fraction ..." are operations and not constructors.

p. 8, r. 311: "T consists of 10 rewrite rules in our formalization." --> "The set of transitions T IS SPECIFIED by 10 rewrite rules in our formalization."

4.2.2 Kripke Structures of Quantum Circuits

There is some confusion about how the quantum states are specified in Maude: first, it is mentioned that
"(qstate: qs) denotes the whole quantum state qs."
and latter
"Each state in S is expressed as {obs}, where obs is a soup of those observable components ....".

Cite this review as
Anonymous Reviewer (2024) Peer Review #1 of "Symbolic model checking quantum circuits in Maude (v0.1)". PeerJ Computer Science

Reviewer 2 ·

Basic reporting

In this submission, the authors consider the problem of model checking quantum circuits against properties specified as formulas in Linear-time Temporal Logic (LTL). Differently from other works in literature, the authors propose a symbolic approach implemented in Maude, so that they can check whether the quantum circuit behaves as expected by manipulating quantum bits and gates in a symbolic way by means of the laws of quantum physics and the rewriting logic underlying Maude. The authors show that their approach is effective by analyzing several quantum algorithms from literature; in particular, they find that the quantum gate teleportation circuit as originally proposed doesn't work as intended and they provide a fixed version that indeed satisfies the desired LTL formulas.

I found the paper reasonable and quite clear to read, and I think its proposed symbolic approach to the verification of quantum circuits is interesting and rather useful. There are however some aspects that don't convince me completely and I think the authors should motivate/clarify better them.

I don't know whether it is a problem with the format required by PeerJ, but the current style of the citations is really annoying, since there is no indication that "Surname (year)" is a citation instead of just regular text. If this is the expected format, the authors make sure that the resulting text is properly readable. For instance, at line 29, "such as Shor's fast algorithms for discrete logarithms and factoring in 1994 Shor (1994)" should be changed to "such as the fast algorithms for discrete logarithms and factoring by Shor (1994)", or in line 45 "There is a symbolic approach Shi et al. (2021)..." that should become "There is a symbolic approach proposed/introduced/presented by Shi et al. (2021)..."

In its current form, it is unclear whether the third contribution (cf. lines 99-102) is from the current draft or it is already present in the SEKE version. Since it should be from the current draft, the authors should make sure the tense of the verbs reflects that this is a contribution of the current version.

The part at lines 183-196 is rather unclear, as it gives no connection with the components of a Kripke structure or an actual example of what these name-value pairs or soups are. The authors should improve it, by providing more explanation or even better an example, for instance based on the superdense code circuit given in figure 1.

In Section 5, it is not so easy for the reader to match the symbols used in the figures with the explanation: while gates like H and X are explicitly named, the gate CX is used but the reader must already know that a CX gate is represented by (+) on the control bit and a full dot on the target, or that a measurement is depicted by a barred arc with an outgoing double arrow pointing to the outcome bit. I think the authors should make this clear, by providing a figure with all operations and relative graphical representations.

Experimental design

no comment

Validity of the findings

On the one hand, quantum circuits have inherently finite behavior, with the possible number of steps given by the number of gates that are used in the circuit. On the other hand, LTL is a logic for expressing behaviors in infinite runs, in particular regarding liveness properties; in the paper, the authors only consider simple reachability properties, for which LTL and the associated model checking algorithms can be considered as overkill. I think the authors do not motivate enough why they have chosen LTL to express properties of systems with finite and bounded executions; the only possibility is that they already have an extension to quantum programs ready, since quantum programs allow for loops (and thus liveness properties can be of interest). In the current draft, however, quantum programs are just considered as possible future work, so they cannot be used as justification for verifying circuits against LTL formulas.

Still about the LTL formulas, the authors use formulas that express properties about the state at termination (or, more precisely, after the quantum gates have applied their operations on the input qubits and then the state is just repeated indefinitely). These formulas usually formalize the expected value of the quantum/classical bits and the formula is satisfied when the circuit works as intended. However, if I look at the stated properties and the corresponding formulas, I see a mismatch between them: take for instance the quantum teleportation algorithm and the statement "At the end, Bob will have |ψ⟩ and Alice will not have anymore". If I look at the formula, only the part "At the end, Bob will have |ψ⟩" is checked, but nothing is said about "Alice will not have anymore". I think it would be better to have formulas that are aligned with the desired properties.

The authors say that the original quantum gate teleportation circuit violates the desired properties and then provide a fixed version. To show that the original version is broken, they do not use the original circuit but a version claimed to be equivalent by the authors that uses the quantum gates available in the proposed Maude framework, since the original gate teleportation uses gates that are not available in the framework. Since the verified circuit is not the original one, in theory it is possible be that its misbehavior is due to the use of different gates rather than by a wrong circuit. I think it would be better if the authors would verify the original version of the circuit, by supporting the Bell measurement in their framework. This would have two advantages: first, the claim that the original circuit is not working as expected would be much more convincing, as the claim is directly supported by the result of the model checking task; second, it would show that it is reasonably simple to extend the proposed Maude framework with new gates.

Additional comments

Minor remarks:
Figure captions should follow the capitalization of normal text, not that of section headers
64: something is strange with the citation "M. Clavel, et al. (2007)" as it doesn't match the other citations (cf. 69: "Do et al. (2022b)")
106: do not use "describes" three times in a row, use a synonym instead
108,258: "make a symbolic model checking" seems incomplete, either "make a symbolic model checker" or "make a symbolic model checking algorithm"
127: "is called as the" -> "is called the"
128: "such as" -> "such as the"
129: missing "."
134-137: better to use the same order for the operators and their matrix representations
147: better to clarify here what ⟨ψ| stands for, instead of in line 212
174: "An" -> "Given an"
200: "as Pakyin et proposed in Pakyin et al. (2017)" -> "as proposed by Pakyin et al. (2017)"
213-214: use bold font for "j", "k", "0", and "1"
224: move "is" just after "Coq"
246,248: better to indent "+" and have it vertically aligned with the "(" in the previous line
467: missing space
528: "not have" -> "not have it"
546,619,699: remove "of scalars"
560 and other lines: try to avoid line breaks in unexpected places
563,642,724: no need to specify what <> represents (even if you just copy/pasted/adapted the content of the different subsections)
594-595: "X, Z, Z" doesn't match "c0, c1, c2" as controlling bits; change one of the two sequences to match the other
708,709: "Sate" -> "State"
743: "qubits" -> "qubit"
780: "Table ??"
795: "constants of complex numbers" -> "constants representing complex numbers"
796,912: "concrete values of real numbers" -> "concrete real number values" or "concrete values for real numbers"
799,901,922,942,947,961,971: "fraction" -> "division"
841: don't use "et al." when there are only two authors
854,855: don't use scalable delimiters for these kets with superscripts
921: clarify what a ctor attribute represents
937: "defined" -> "introduced"
987: this equation seems wrong, as I would expect to have Sqrt(- N) = i .* sqrt(N)
References: make sure capitalization is correct by using {}, e.g., "{Einstein}-{Podolsky}-{Rosen}" in the bibtex for line 1011
1062: do not hide other authors behind "et al."

Cite this review as
Anonymous Reviewer (2024) Peer Review #2 of "Symbolic model checking quantum circuits in Maude (v0.1)". PeerJ Computer Science

Reviewer 3 ·

Basic reporting

This area is basically fine. My full review (section 4) includes some requests to better situate the work within the field.

Experimental design

This is fine.

Validity of the findings

The research findings are clearly reported.

Additional comments

The paper presents the use of the Maude language and model-checking system to model and verify quantum communication protocols. It contributes to a research topic that has been going on since the mid-2000s, on the application of formal methods to quantum systems.

Among previous work on this topic, some has attempted to use standard tools such as PRISM, and some has developed specialised tools. Some work has aimed at fully automatic analysis, and some is based on interactive theorem proving and requires human input. The interesting aspect of this paper is that it uses an existing automatic system, Maude, and finds a way to represent quantum states and operations in terms of Maude's term rewriting framework, thus avoiding the need to implement a separate representation of quantum states (e.g. as was attempted in early work by Gay and Nagarajan with PRISM).

The paper shows that this approach in Maude can be used to analyse several simple quantum communication protocols. This is essentially at the same scale/complexity as previous work, although the final case study on quantum network coding is a little larger with ten qubits.

In my view, the paper makes an interesting and worthwhile contribution to the topic of formal methods for quantum systems. Basically it is worth publishing.

I would like to see some revisions, in several categories.

1. There are a number of detailed technical aspects, described below for each section, that need to be clarified.

2. The introduction motivates the study of quantum computing by describing quantum algorithms and quantum cryptography, but the paper doesn't address either of these topics - it focuses on quantum communication protocols. The actual content of the paper is clear from the abstract, but I would like the authors to revise the introduction in order to bridge the gap between the discussion of quantum algorithms and quantum cryptography, and the quantum communication protocols that they actually analyse.

3. There is another gap between the informal description of the quantum protocols, which include communication between two or more participants, and the formal models, which are expressed in terms of quantum circuits. This is a partly-hidden step, before the formal modelling starts: expressing the protocol as a circuit already abstracts away from the details of communication and concurrency. There are potential communication and concurrency errors in protocol implementations that this approach to modelling cannot discover. The authors should explain what they are doing, and note that this is not the only approach to modelling quantum protocols (e.g. the work of Ardeshir-Larijani, Gay and Nagarajan includes explicit modelling of communication and concurrency).

4. The topic of formal methods for quantum computing began almost 20 years ago, and as far as I know, all of the published work has used case studies at the same simple level with a small number of qubits. Meanwhile there have been huge developments in practical and commercial quantum computing systems, such that the big players are now working towards systems with hundreds or thousands of qubits. It's clear that the kind of model-checking approaches described in this paper (and previous work) can't scale to hundreds of qubits if systems are modelled in a monolithic way. Techniques for modular analysis will have to be developed. The authors should at least acknowledge this and make some meaningful suggestions towards modular analysis of larger systems.



Detailed comments on each section:


Abstract

This is a clear and concise summary of the approach and results of the paper.


1. Introduction

This clearly sets out the aims and approach of the paper, motivated by the growth of interest in quantum computing.

There is, however, a gap between the examples of quantum computing and quantum cryptography quoted for motivation, and the simple quantum communication protocols used as case studies. Can your approach deal with algorithms such as Grover or Shor? Can it deal with cryptographic protocols with an attacker? Can it deal with error correction with a source of noise?

The additions to this version of the paper, with respect to the original conference paper, are clearly stated.


2. Preliminaries

The essentials of quantum computing, Kripke structures and LTL are clearly defined.


3. Symbolic reasoning

This is reasonably clear, but I have some questions.

- You explain that Maude supports rational numbers and that you extend this to support complex numbers. But you are also adding some irrational numbers because you have a square root operation. So, exactly which number field do you end up with? I don't think all complex numbers can be expressed using finite syntax based on rational numbers and square roots, for cardinality reasons. So, for example, can you represent (or approximate) the states that occur in Grover's or Shor's algorithm?

- You say that your equations are sound, but what about completeness? Are there protocols whose correctness cannot be checked in your system because some necessary equalities cannot be established from your axioms? I wonder whether this also applies to the square roots as well as to the matrix operations.

- At the end of the section you say that Maude can do symbolic reasoning about matrices instead of explicit calculation. But is this symbolic approach more efficient in general? Representing a 2x2 matrix as a sum of four little outer products is essentially the same as representing it as a two-dimensional array. This leads to a more general question about efficiency. The experimental results are for small protocols with few qubits, so the timings are fairly short. But I think the model is based on an explicit state-space representation, so it is to be expected that the time taken to just simulate the quantum operations (regardless of model-checking) will grow exponentially with the number of qubits in general. Please comment on this, and whether there are any possibilities for more efficient simulation, e.g. taking advantage of the entanglement structure of the quantum state. There is a lot of literature on efficient quantum simulation, and model-checking is simulation plus property checking.


4. Formal specification

I'm confused about the representation of quantum states. 4.2.1 describes single qubit states and multi-qubit entangled states, giving the impression that a quantum state is a collection of components in which each component is either a single qubit or a collection of entangled qubits. I don't see how the distinction between single and entangled qubits is maintained as quantum operations are applied. It might be useful to do that for efficiency, but you would need a way of detecting when an entangled part of the state has become separable. Please clarify exactly what the representation is, and how the quantum state is manipulated by operations.

Please be clear about whether or not you have implemented a set of gates that is sufficient for universal quantum computation. The gates listed from lines 288 to 295 are not universal, but then you list more gates including phase (S), pi/8 (T) and doubly controlled gates, so that should give you a universal set. This is related to the question of whether you can represent the states and operations needed for e.g. Shor's algorithm. If you have a universal set of gates then you should be able to implement algorithms as well as the simple protocols used in your case studies.


5. Symbolic model checking

5.1 Superdense coding

There's a mismatch between the informal description of the protocol and the circuit model, which is worth explaining. The circuit doesn't represent Alice and Bob explicitly, so there is nothing in it corresponding to "the entangled state is shared between Alice and Bob using a quantum channel" (line 413). All the other protocols / models in the paper have the same feature of flattening a communication protocol into a circuit.

"Alice needs to send two classical bits" (line 415) is misleading: there is nothing in the circuit corresponding to sending two bits; sending two classical bits is the overall effect of the protocol, not a step within the protocol.

It's disappointing that the four possible two-bit values have to be represented separately in the Maude model, and the model-checker called four times. This is a constrast to the analysis of teleportation, where we see the benefit of symbolic model-checking in allowing a single model-checking run to check the protocol for all possible input qubit states. In dense coding, is it possible to treat the classical two-bit value in a similar way, as an input to the protocol rather than a parameter, so that you can also take advantage of symbolic model-checking here? If this is not possible, can you represent the four possibilities within a single model by using non-deterministic or probabilistic transitions from a single initial state to the four states corresponding to the four two-bit values?

Why are the specifications expressed with Prob > 0 ? Surely you want the protocol to reach the correct end state with probability 1 ?

Here and in the other case studies, you say "no counterexample is found in just a few moments". I would leave comments on the timing to the quantitative results - "just a few moments" is vague and too informal.

It would be worth commenting that when writing your specifications, you only use a very limited kind of LTL formula that enables you to compare parts of the initial and final states.


5.2 Quantum teleportation

Again, why does the specification refer to a non-zero probability, rather than probability 1?

Referring to qubitAt(Q,2) only makes sense if qubit 2 in state Q is not entangled with other qubits, so how is this checked? Actually the fact that the final qubit is not entangled with any other qubits should be part of the informal specification of the protocol, so it needs to be included in the formal specification somehow.


5.3 Quantum secret sharing

I have the same question here about non-zero probability (why is it not probability 1 ?). Also the same question about how you check that the final qubit is not entangled with other qubits.

You explain that the purpose of the QSS protocol is that Bob and Charlie can cooperate to obtain Alice's qubit, but neither Bob nor Charlie can independently obtain Alice's qubit. I think you have model-checked the first property, but not the second. Is there any way you can model general behaviour of Bob, say, to try to obtain the qubit, and check some upper bound on the probability of success? Ideally, you would be able to write a specification of QSS that is not satisfied by simply teleporting the qubit from Alice to Charlie. If this is not possible, please explain that you have only formalised and verified one aspect of QSS.



5.4 Quantum gate teleportation

Here I have the same question about the non-zero probability.

Using qubitAt(Q, 3 2) in the specification is correct because it considers the combined state of two qubits, to allow for the possibility that they are entangled. But as before, you also need to know that the state of these two qubits is separable from the rest of the quantum state.



6 Remark on quantum gate teleportation

This is a nice demonstration of your approach - you found a mistake in a published circuit and you were able to correct it.


7 Experimental results

Again, I wouldn't say "just a few moments" (line 785). You are using the same informal phrase to describe times from 1ms to 2446ms, which span three orders of magnitude!

You say that you want to do case studies with more qubits. Considering that your running times seem to be increasing rapidly with the number of qubits, and given that you are using an explicit state representation, it seems clear that model-checking times with your approach are going to increase exponentially with the number of qubits. You can easily check this by generating random circuits and running your model-checker with a trivial specification. Given that, it would be worthwhile to comment on how a model-checking approach that can only handle a limited number of qubits can be useful in the context of verifying a larger system. (I believe it can, through modular analysis of key components).


8 Limitations

You say that you only support a limited set of quantum gates, but I think CNOT, H, S and T give a universal set in the sense of being able to approximate any quantum operation. So the question of whether you can express all protocols would be a question of how much encoding is required if the protocols use gates that you have not directly implemented.

I don't think the restriction to the standard basis is really a limitation, because for measurements in other bases you can always do a basis transformation followed by a standard basis measurement.


9 Related work

You discuss some work by Gay and Nagarajan, but more recent work by Ardeshir-Larijani, Gay and Nagarajan (TACAS 2014, ACM TOCL 2020) models quantum protocols and explicitly includes classical and quantum communication between concurrent processes, instead of treating protocols as circuits. Can you use Maude to also model the concurrent behaviour in quantum protocols?

It would be worth including a comparison with the Quantomatic system, which is a rather different approach to automatic analysis of quantum systems based on graph rewriting.




Minor points


Bibliography entries not fully specified (missing publication details):

Gay, Nagarajan and Papanikolaou 2005

Gay, Nagarajan and Papanikolaou 2007

Joy, Sabir, Behera and Panigrahi 2018

Nagarajan and Gay 2002

Ying and Feng 2018


In general, citation keys within the text are not formatted correctly. Either the entire citation key should be in parentheses, or the name of the author should be used in a grammatically correct way within the sentence.

Cite this review as
Anonymous Reviewer (2024) Peer Review #3 of "Symbolic model checking quantum circuits in Maude (v0.1)". PeerJ Computer Science

---

## Round 0.2 · Major Revisions

As you can see, all 3 reviewers still have substantial comments and suggestions for improvement. They find the work interesting, but it is imperative that all their concerns are addressed before I would be able to accept the paper.

Reviewer 1 ·

Basic reporting

The authors give detailed responses to the reviewers' remarks and most of these remarks
are satisfactorily addressed in the paper.

Experimental design

The current version of the Maude specification of quantum circuits is not quite trivial and it is a promising step towards a more elaborated support for quantum computing.

Validity of the findings

From my point of view, there are still some aspects that should be
accordingly addressed.

Regarding the response related to the concept "general framework to
formally specify and verify quantum circuits":

The current development is just a "Maude-based support for analyzing quantum circuits".
This can be a first step toward a general framework, but for that, the paper should include a
section formally describing 1) What is the rewrite theory associated with a quantum circuit
(via Kripke structures), 2) What is the relationship between the two concepts, and 2) How
the properties of quantum circuits can be represented as a system specification.
In the current version of the paper, this is described only at the level of implementation.
Describing it at the level of concepts is helpful for both readers' communities:
quantum computing community can see what they gain using the Maude-based support, and the
rewriting logic community can see how Maude can contribute to the specification and
analyzing the quantum circuits/programs.
Moreover, it could be a strong argument for the soundness of the approach.

I am not sure that the term "eventual properties" is the most appropriate for the
"class of liveness properties, to express the desired properties
for several quantum communication protocols". Moreover, this (sub)class of properties
is not formally defined in the paper and it is not argued why this class is interesting for
the specification of quantum circuits.

It will be helpful to see examples of quantitative resp. qualitative properties of quantum
circuits; the definition from the footnote (page 3) is too general.

Regarding the response related to Quantum Gate Teleportation:
"In short, in their
original quantum circuit, they used the Bell measurement without explicitly specifying the circuit for it. Based
on their description and the Bell measurement, we constructed a quantum circuit for the Bell measurement,
thereby creating a possible quantum circuit for the protocol."
It seems to be a sub-specification problem with the original definition and not that it "does not satisfy its desired property".

The authors claim in several places that the Maude theories are generic and can be reused such that
"If readers are interested in
extending it as in this case, they can do so without any difficulties.".
it would be helpful for readers to include in Section 8 a short description of how these theories can be reused.

Additional comments

Detailed comments:

In some places, introducing only "by" does not solve the readability for citations. E.g.,
on page 2, rows 74-78,
"Superdense Coding by Bennett and Wiesner (1992), Quantum Teleportation by Bennett et al. (1993),, ..."
could become
"Superdense Coding, introduced by Bennett and Wiesner (1992); Quantum Teleportation, introduced by Bennett et al. (1993); ..."

p5, r179: "A Kripke structure K is ⟨S, I,T,A,L⟩" --> "A Kripke structure K is a tuple ⟨S, I,T,A,L⟩"

Section 4: change
"formalize" into "specify in Maude"
and
"Formalization" into "Maude Specification".
The quantum circuits are already formally defined.

Section 4.2.1:
Explain what exactly means qi, qi, . . . ,qj, ...

p9, r320:
"...describe universal quantum computation, however, we need.."
-->
"...describe universal quantum computation. However, we need..."

p11, r433-434:
"However, we intend to describe the protocols as if participants communicate with each other in the protocols."
is too vague, please elaborate a bit.

p25, r1007:
It is a bit strange to define "errC" as a complex number. Why don't use a sort "MaybeComplex" with
"Complex < MaybeComplex" and define "errC" of sort "MaybeComplex" (which is similar to the maybe/option types).

For instance, on r1042, there is the equation
eq C .* 0 = 0 .
If C is errC, then this equation is not correct.

Cite this review as
Anonymous Reviewer (2024) Peer Review #1 of "Symbolic model checking quantum circuits in Maude (v0.2)". PeerJ Computer Science

Reviewer 2 ·

Basic reporting

I think the authors have improved the draft, compared with the previous submission. There are still few aspects that need to be further ameliorated to be suitable for publication.

Regarding the case studies, I think they would be more convincing if the authors would check more properties, with them being not satisfied when expected. For instance, in the Superdense Coding, I think it would be meaningful to have e.g. a check of "red modelCheck(init0, gateXProp) ." with the corresponding failure: this would complement the current positive results since it would confirm that Bob cannot receive bits that differ from the ones sent by Alice. Clearly checks should be performed on all combinations of initj and gateWProp, with just a summary provided.
Similarly, it would be nice to see that in the Quantum Teleportation the property defined as in lines 585-588 but with the =/= replaced with == in line 587 is not satisfied, i.e., indeed Alice keeps her initial qubit with probability 0.

Still on the case studies, the authors propose properties that are all qualitative (cf. the qualitative fragment of PCTL for Markov chains, Def. 10.42 of Baier-Katoen: "Principles of Model Checking"). Since the authors say that they support quantitative properties, it would be nice to see at least one of such formulas in the experiments.

Regarding the experimental results, I would say that the proposed approach is not a solution for the large number of rewriting steps, but it is the cause of them. Maybe a different encoding/solution method can tackle systems with the same number of states with much fewer rewriting steps (or even without any rewrite), but this cannot be known until such a different approach is proposed. So I would not use the number of rewriting steps as a justification for the usefulness of the proposed approach.

I am still not convinced about the presentation of soups in Section 2. Since they are only used briefly in Section 4.2.2, I would say that soups should be completely removed from Section 2, and just used in Section 4.2.2 without even needing a name there. Just say that a state is formed by one of each of the 5 observable components.

Some sentence still need improvement with respect to the occurrence of citations in the text, like lines 64, 69, 70, 179, 214.


Now, I want to reply to some response by the authors about some points I raised in my previous and that I am not so satisfied by their response.

#3: My comment can be split into two parts, one about the temporal operators and one about soups and name-value pairs, with this second part explicitly mentioned; the authors' response completely misses the second part, that I now suggest above to remove. Regarding the first part, the current text at lines 200-201 should be moved just after the LTL grammar, as such text is about syntax and syntactic sugar for formulas; it is not relative to the LTL semantics. Clearly the remaining part of the text at lines 199-205 needs to be made smooth again.
#5: I agree that LTL permits to specify interesting properties that go beyond reachability. Still, for the current paper that is about quantum *circuits*, I think the full LTL with infinite semantics is overkill, since the systems have only finite paths and the authors have to make them artificially infinite by cycling forever after the last gate has been applied. So, I still think the use of LTL needs some more justification, e.g. by analyzing a non-trivial property that is not a reachability property. As a note, LTL has no requirement about the fact that a system is finite or infinite, but LTL only cares about systems with infinite paths/runs; a system may be infinite, but with all paths of finite length so none of them would be considered in the LTL semantics.
#9: The issue with "M. Clavel, et al. (2007)" is not about the missing "b" in the year, but in the format of "M. Clavel, et al.". The issue is now solved as a byproduct of correcting the corresponding bibtex entry to have all authors
#11: the equation at line 987 in the original submission is wrong. Period. It is: "eq Sqrt(- N) = i .* N .". The current version is the expected one: "eq Sqrt(- N) = i .* Sqrt(N) ." even if it has been corrected without marking it as changed.
#12: the response doesn't match the issue, even if the entry has been corrected

Experimental design

no comment

Validity of the findings

no comment

Additional comments

Minor remarks:
34: "mechanical" -> "mechanics"
55: provide some more details about this gap
95: no need to reintroduce the acronym LTL
106: "conduct" -> "verify"/"analyze"
116-125: since you use commas inside the description of the single sections, use semicolon as separator between the sections
Tables 1,3: caption goes above the table
145-146: "transformation U can ... matrix such" -> "transformation can ... matrix U such"
179: "K is" -> "K is a tuple"
180: missing Oxford-comma before "and" (for uniformity with other lists)
230: "are the" -> "are the ones of the"
231: "scalar multiplication of matrices ·" -> "scalar multiplication ·"
235: use bold font for 0 and 1 in "j, k ∈ {0, 1}"
Table2,L1: <1|1> should be <1|0>
255-270: it would be helpful to explain each step by indicating the applied law/motivation for the simplification
273: use bold font for i and j in "<i|j>"
326-330: do no use "denotes" 5 times; use synonyms
337: "transition" -> "transitions"
449: the H gate is applied on q0, not q1, according to Figure 2 and the encoding in 5.1.2.
452: "expected as" -> "expected to be"
520: "that if" -> "that"
772-774: the presentation needs some improvement, so to clarify that for |00>+|11> there are two outputs, i.e., 00 and 01, both occurring with the same probability; similarly for |01>+|10>
874: "formation" -> "formulation"
875: "quantum" -> "quantum states"
899: "multiple-dimensional" -> "multi-dimensional"
904: "of" -> "of the"
938: "XZ" -> "ZX"
954: "we" -> "they"
988: "constructors" -> "operations"/"operators"
1000: "concrete values of real numbers" -> "concrete real number values" or "concrete values for real numbers"
1009: something is missing here, like a verb between "ctor attribute" and "the constructors"
1028-1029: since you use commas inside the description of the single variables, use semicolon as separator between them (e.g., "Nat, R, R1" -> "Nat; R, R1")
1040,1049,1090: "are" -> "is"
1050: shouldn't this require the condition "C =/= 0"? and similarly the following lines
1075: why the square root of negative values is defined only for the naturals, and not also for rationals/reals?
1137,1145,1162,1180: incorrect capitalization (Bell, Markov, IBM, Coq)
1193: missing conference name

Cite this review as
Anonymous Reviewer (2024) Peer Review #2 of "Symbolic model checking quantum circuits in Maude (v0.2)". PeerJ Computer Science

Reviewer 3 ·

Basic reporting

No comment

Experimental design

No comment

Validity of the findings

No comment

Additional comments

My response to the authors' changes.


Concern 1

You have added the text "In this paper, although we do not directly tackle quantum circuits for complicated quantum algorithms..."

This clarifies that your case studies are protocols not algorithms, as I requested.

However, you write "our approach can be extended to formalize and verify them provided that necessary quantum gates are developed and our symbolic reasoning for complex numbers is enhanced to be able to describe and reason about their behaviors adequately"

Here you are asserting that you can extend your approach to handle algorithms such as Shor and Grover. This is far from clear. The statement is too strong. Please revise this to clarify that extending your approach to handle these algorithms would require further research.


Concern 2

You have added "To handle quantum cryptography, such as BB84 by H. (1984) and B91 by Ekert (1991), our formalization needs to be extended to explicitly express concurrency and communication among participants in quantum protocols."

I believe this would be a very significant extension of your formalisation, requiring substantial extra research. Please revise the text to clarify that.


Concern 3

This has been resolved.


Concern 4

This is very similar to Concerns 1 and 2, which I have commented on.


Concern 5

This has been resolved.


Concern 6

This has been resolved.


Concern 7

Maybe my comment was not clear. You give the example of the X gate, which in Dirac notation is a sum of two outer products: |0><1| and |1><0|. This is in comparison to a 2x2 matrix representation, which includes the 0 entries. So it seems that Dirac notation gives you some advantage in representing a sparse matrix. On the other hand, applying operators in Dirac notation requires symbolic rewriting instead of simply matrix multiplication. And if you consider the Hadamard operator, it isn't a sparse matrix, so it's not clear that there is any advantage in using Dirac notation. So I don't think it's obvious that Dirac notation has the efficiency benefits that you claim. Please comment on this.


Concern 8

I don't think my comment was clear. What I am asking about is the following. If I understand correctly, your representation of quantum states shows that some groups of qubits are entangled while some qubits are not entangled. I didn't see where you take advantage of this aspect of your representation. What benefit does it give you to represent groups of entangled qubits, rather than treating the whole state uniformly? And the second part of the question was how you detect that qubits can become disentangled. Your response says that you have some heuristics for that. Anyway, I don't think you have revised the paper to deal with this comment, so please do that.


Concern 9

This has been resolved.


Concern 10

This has been resolved.


Concern 11

Please revise the paper to explain why the input to superdense coding cannot be treated as a two-bit value.


Concerns 12, 15 and 17

I didn't understand your answer. There are some quantum protocols or algorithms in which there is a certain probability < 1 of success, and the algorithm might have to be executed repeatedly in order to produce the answer. But teleportation succeeds with probability 1. Specifically, there is a measurement as part of the protocol, with four outcomes that have probability 1/4 each. In each case a different correction is applied in order to make the final state correct - so summing the probabilities, the probability of success is 1. We can imagine an incorrect teleportation protocol in which there is an extra measurement at the beginning, such that with probability 1/2 the protocol stops and does not complete the teleportation, and with probability 1/2 it continues to complete the standard teleportation protocol. Can your specification detect the difference between these two protocols?


Concern 13

This has been resolved.


Concern 14

This has been resolved.


Concern 16

I didn't understand your answer to this point.


Concern 18

This has been resolved.


Concern 19

This is a repeat of Concerns 12 and 16, which I am not yet convinced about.


Concern 20

This has been resolved.


Concern 21

This has been resolved.


Concern 22

This has been resolved.


Concern 23

My point here is that you seem to be unfair to yourselves: there is no need to represent states in an alternative basis, because if you want to measure in a different basis, you can just use a unitary transformation to change basis, do a standard basis measurement, then change basis again.


Concerns 24 to 27

These have been resolved.

Cite this review as
Anonymous Reviewer (2024) Peer Review #3 of "Symbolic model checking quantum circuits in Maude (v0.2)". PeerJ Computer Science

---

## Round 0.3 · Minor Revisions

The reviewers find that your paper has improved, and deserves publication. However, they also still have suggestions for possible improvements. Please do take these into account when preparing a new version of the paper. In particular, the points raised by reviewer 1 should be addressed in a more convincing manner.

Reviewer 1 ·

Basic reporting

The authors tried to respond as best as possible to all the comments of the reviewers.

It seems that this piece of work is part of a bigger project, where the authors intend to use the rewrite logic and the Maude system to analyze Quantum circuits. This is a basic step toward accomplishing this project and I think it deserves to be published.

Experimental design

As I said in a previous review, the Maude specification of quantum circuits is not quite trivial. It has some promising potential, but at the same time, it seems it has some significant limits.

Validity of the findings

Most of the answers are satisfactory, but some of them are not convincing.

The description of the relationship between QC spec and the associated Rewrite theory is still very informal and given in general terms. From this, it is hard to formally conclude the soundness of the approach. However, perhaps a more formal approach is hard to find.

Regarding the answer: "... where something good eventually happens. For example, a qubit at the
final state for each possible execution path is the same as another qubit at the initial state with a non-zero probability.".
I am not sure that this is a good example. Note that "A property that gives a specific bound to the "good thing" is a safety property".

Regarding my previous comment on "how these theories can be reused": the authors added a very general description and I do not know how much it helps a user to reuse the theories described in the paper.

Additional comments

The initial Maude theory for the complex numbers included some errors, detected by reviewers. This is mentioned as a contribution. How can we know that it is accurate now?

Cite this review as
Anonymous Reviewer (2024) Peer Review #1 of "Symbolic model checking quantum circuits in Maude (v0.3)". PeerJ Computer Science

Reviewer 2 ·

Basic reporting

I think the paper is in a suitable shape to be accepted; I have no other requests except for few remaining minor remarks that can be managed while preparing the final version.

Experimental design

no comment

Validity of the findings

no comment

Additional comments

Minor remarks:
45-47: I would move this sentence at the end of the paragraph
76-80, 441-451: do not use "introduced" so many times; use some synonym instead
159: "one" -> "1"
264 or 138: make it explicit that the bra-ket notation means <ψ| and |ψ>
282: "X|1>" -> "X|1> = |0>"
321: "Figure 2, Figure 3, and Figure 4" -> "Figures 2-4"
330,331: "denote" -> "denotes"
340 vs. 359: use punctuation uniformly between similar lists
399 to the end of the paper: "in qstate observable component" -> "in the qstate observable component" and similarly for other observable components and other prepositions (e.g., when) or without prepositions
495 to the end of Section 6: notation can be improved by adding spaces between the single gates (e.g., "M(0) M(1)" instead of "M(0)M(1)"); cf. 621
590: "init1." -> "init1,"
725: "the H on" -> "H on" or "the H gate on"; similarly at 805
803: "Four" -> "Fourth"

Cite this review as
Anonymous Reviewer (2024) Peer Review #2 of "Symbolic model checking quantum circuits in Maude (v0.3)". PeerJ Computer Science

---

## Round 0.4 · accepted · Accept

Congratulations, the reviewers approve your changes and are happy to accept the paper now.

Reviewer 1 ·

Basic reporting

My option is to accept the current version for publication.

Experimental design

no comment

Validity of the findings

no comment

Cite this review as
Anonymous Reviewer (2024) Peer Review #1 of "Symbolic model checking quantum circuits in Maude (v0.4)". PeerJ Computer Science